# Dengue virus preferentially uses human and mosquito non-optimal codons

Luciana A Castellano [ID][1], Ryan J McNamara[1], Horacio M Pallarés [ID][1,2], Andrea V Gamarnik[2], Diego E Alvarez [ID][3] & Ariel A Bazzini [ID][1,4 ✉]

## Abstract

Codon optimality refers to the effect that codon composition has on messenger RNA (mRNA) stability and translation level and implies that synonymous codons are not silent from a regulatory point of view. Here, we investigated the adaptation of virus genomes to the host optimality code using mosquito-borne dengue virus (DENV) as a model. We demonstrated that codon optimality exists in mosquito cells and showed that DENV preferentially uses nonoptimal (destabilizing) codons and avoids codons that are defined as optimal (stabilizing) in either human or mosquito cells. Human genes enriched in the codons preferentially and frequently used by DENV are upregulated during infection, and so is the tRNA decoding the nonoptimal and DENV preferentially used codon for arginine. We found that adaptation during single-host passaging in human or mosquito cells results in the selection of synonymous mutations towards DENV's preferred nonoptimal codons that increase virus fitness. Finally, our analyses revealed that hundreds of viruses preferentially use nonoptimal codons, with those infecting a single host displaying an even stronger bias, suggesting that host–pathogen interaction shapes virus-synonymous codon choice.

**Keywords** Codon Optimality; Dengue; Mosquito; tRNA; Viruses
**Subject Categories** Microbiology, Virology & Host Pathogen Interaction; RNA Biology

## Introduction

Dengue is the most prevalent mosquito-borne viral disease in the world, causing an estimated 390 million infections each year (Bhatt et al, 2013). This disease is caused by four (DENV1-4) distinct but genetically related serotypes. DENV is a single-stranded positive-sense RNA virus that is transmitted to humans by infected *Aedes* species (*Ae*. Aegypti or *Ae*. Albopictus) mosquitoes (Henchal and

Putnak, 1990). Besides the strict dependence on the host translation machinery to synthesize their protein components, RNA viruses display compact genomes with a limited coding capacity and have evolved to adapt the host cell translation resources (Ahlquist, 2006). Beyond the amino acid composition of viral coding sequences, selective pressures acting on virus genomes determine their particular nucleotide composition (Fros et al, 2021; Kustin and Stern, 2021; Sexton and Ebel, 2019). On the one hand, the presence of cis-acting regulatory sequence and structural motifs often pose a constraint to mutation as changes in these elements can be detrimental or even deleterious (Fros et al, 2021; Gumpper et al, 2019; Sexton and Ebel, 2019). On the other hand, as the redundancy of the genetic code allows most amino acids to be specified by more than one synonymous codon, selection leads to various forms of compositional bias in viral genomes, including nucleotide bias (Balzarini et al, 2001; Berkhout et al, 2002; Jenkins et al, 2001; Kapoor et al, 2010; Lobo et al, 2009; Muller and Bonhoeffer, 2005; Shackelton et al, 2006; van der Kuyl and Berkhout, 2012; van Hemert and Berkhout, 2016; van Hemert et al, 2007), codon bias (Adams and Antoniw, 2004; Bahir et al, 2009; Belalov and Lukashev, 2013; Berkhout et al, 2002; Bouquet et al, 2012; Butt et al, 2014; Cai et al, 2009; Chen, 2013; Fu, 2010; He et al, 2017; Jenkins and Holmes, 2003; Jenkins et al, 2001; Kumar et al, 2016; Li et al, 2012; Liu et al, 2010; Nougairede et al, 2013; Plotkin and Dushoff, 2003; Tao et al, 2009; van Hemert et al, 2007; Wong et al, 2010; Zhao et al, 2003; Zhong et al, 2007), dinucleotide bias (Antzin-Anduetza et al, 2017; Atkinson et al, 2014; Di Giallonardo et al, 2017; Gaunt et al, 2016; Kunec and Osterrieder, 2016; Shackelton et al, 2006; Simmonds et al, 2015; Tao et al, 2009; Tulloch et al, 2014; Upadhyay et al, 2014; Washenberger et al, 2007; Witteveldt et al, 2016) and codon pair bias (Coleman et al, 2008; Gao et al, 2015; Le Nouen et al, 2014; Leifer et al, 2011; Li et al, 2018; Martrus et al, 2013; Mueller et al, 2010; Ni et al, 2014; Wang et al, 2015; Yang et al, 2013) that are often related to host tropism. For example, CpG dinucleotides were shown to be underrepresented in vertebrate viruses, while invertebrate viruses lack this CpG suppression (Gaunt and Digard, 2022; Simmonds et al, 2015; Simmonds et al, 2013). While many studies have proposed the idea that codon bias in viral genomes may be selected as a strategy for regulating the translation of viral proteins (Bahir et al, 2009;

[1]Stowers Institute for Medical Research, 1000 E 50th Street, Kansas City, MO 64110, USA. [2]Fundación Instituto Leloir, Instituto de Investigaciones Bioquímicas de Buenos Aires IIBBA-CONICET, Ciudad Autónoma de Buenos Aires, Argentina. [3]Instituto de Investigaciones Biotecnológicas, Universidad Nacional de San Martín-CONICET, San Martín B1650, Argentina. [4]Department of Molecular and Integrative Physiology, University of Kansas Medical Center, 3901 Rainbow Blvd, Kansas City, KS 66160, USA. ✉E-mail: arb@stowers.org

Belalov and Lukashev, 2013; Berkhout et al, 2002; Chen, 2013; Cristina et al, 2015; Jenkins and Holmes, 2003; Jenkins et al, 2001; Li et al, 2012; Moratorio et al, 2013; Tao et al, 2009; van Hemert and Berkhout, 2016; van Weringh et al, 2011; Wong et al, 2010), it has also been proposed that viral codon bias is dictated primarily by the nucleotide composition of the viral RNA genome (Atkinson et al, 2014; Burns et al, 2009; Kunec and Osterrieder, 2016; Moura et al, 2007; Tulloch et al, 2014; van Hemert et al, 2016). Untangling these conflicting drivers of evolution has remained challenging (Gaunt and Digard, 2022; Gumpper et al, 2019). For instance, the DENV genome has been subjected to various evolutionary constraints and selective pressures, as shown by the presence of evolutionary conserved cis-acting regulatory motifs and adaptation to the divergent dinucleotide and codon frequencies, along with codon pair preferences exhibited by genes encoding proteins in humans and mosquitoes (Chin et al, 2023; de Borba et al, 2015). Synonymous variations were shown to critically influence the viral life cycle (Chin et al, 2023; Martinez et al, 2016). In this sense, vaccines were generated by viral attenuation due to synonymous mutations, illustrating that synonymous mutation are not silent from the regulatory point of view (Broadbent et al, 2016; Coleman et al, 2008; Fan et al, 2015; Goncalves-Carneiro and Bieniasz, 2021; Le Nouen et al, 2014; Mueller et al, 2010; Nogales et al, 2014; Shen et al, 2015; Wang et al, 2015; Yang et al, 2013).

Synonymous codon can also influence protein abundance across species by a post-transcriptional mechanism known as codon optimality (Bazzini et al, 2016; Boel et al, 2016; Burow et al, 2018; Forrest et al, 2020; Medina-Munoz et al, 2021; Mishima and Tomari, 2016; Narula et al, 2019; Presnyak et al, 2015; Richter and Coller, 2015; Wu and Bazzini, 2018; Wu and Bazzini, 2023; Wu et al, 2019). "Codon optimality" is the mechanism by which mRNA translation affects mRNA stability and translation efficiency in codon-dependent manner (Bazzini et al, 2016; Presnyak et al, 2015; Wu et al, 2019). Codons that enhance mRNA stability are defined as "optimal codons", while "nonoptimal" codons have the opposite effect (Wu et al, 2019) (Fig. EV1A). Thus, mRNAs enriched in optimal codons tend to be more stable and more efficiently translated than mRNAs enriched in nonoptimal codons (Bazzini et al, 2016; Presnyak et al, 2015; Wu et al, 2019). While the molecular mechanisms involved in codon optimality remain poorly characterized, optimal codons are associated with higher levels of tRNA and higher charged to uncharged tRNA ratios (Bazzini et al, 2016; Despic and Neugebauer, 2018; Frumkin et al, 2018; Presnyak et al, 2015; Rak et al, 2018; Richter and Coller, 2015; Wu et al, 2019). Unlike codon usage, which refers to the frequency of each codon in a given transcriptome, "codon optimality" refers to the effects that specific codons have on mRNA stability and translation efficiency (Hanson and Coller, 2018). Frequently, these two terms are employed interchangeably, however codon usage and optimality do not exhibit a strong correlation in vertebrates (Bazzini et al, 2016; Wu et al, 2019). For example, the codon GAA is one of the most frequently used codons in human cells but it is nonoptimal (Wu et al, 2019). While it has been previously shown that vertebrate viruses exhibit codon bias that does not mimic their hosts usage (Castells et al, 2017; Chen et al, 2020; Cristina et al, 2015; Moratorio et al, 2013; Simon et al, 2021), the relationship between viral codon preference and host codon optimality remains unexplored.

Here, we demonstrated that codon optimality exists in mosquito cells and defined its codon optimality code to address DENV codon preference relative to both human and mosquito host codon optimality codes. We also investigated whether host gene expression, including tRNA abundance, changes in a codon-dependent manner upon DENV infection and whether selective pressures act on virus-synonymous codon choice. This work revealed that DENV and many other viruses preferentially use nonoptimal codons compared to their host which has important implications for understanding host–pathogen interactions.

# Results

## Many viruses preferentially use nonoptimal codons relative to humans

To determine which codons DENV preferentially uses relative to its host, a Relative Synonymous Codon Usage (RSCU) fold change was calculated as the ratio of RSCU observed in DENV2 to that of its host (e.g., human) per amino acid (Dataset EV1). RSCU is independent of amino acid frequency (Fig. 1A,B). Then, the correlation between the RSCU fold change and the codon stability coefficient (CSC) from human cells (Wu et al, 2019), a measure of codon optimality, was calculated to investigate the relationship between DENV's codon preference and the hosts' codon optimality. A negative correlation was observed between the RSCU fold change of DENV1 ($R = -0.31$, $P = 0.016$), DENV2 ($R = -0.2$, $P = 0.13$), DENV3 ($R = -0.28$, $P = 0.034$) and DENV4 ($R = -0.26$, $P = 0.047$) and codon optimality (Figs. 1C and EV1B), suggesting that DENV preferentially uses nonoptimal codons relative to humans. For example, AGA is preferentially used by DENV2 to encode arginine relative to humans and is the most nonoptimal codon encoding arginine in human cells (Fig. 1D). A similar trend towards a preferential use of nonoptimal codons was observed for other amino acids, such as isoleucine, glutamine, lysine, and leucine (Fig. 1D). Interestingly, nonoptimal codons unrelated with CpG dinucleotide bias such as ATA, CAA, ACA, AGA, AAA, CCA, TCA, and TTA, are preferentially used by DENV over other synonymous codons which are also unrelated with CpG dinucleotide (Fig. 1D). As a control, no significant correlation was observed between the RSCU fold change of the DENV2 3' untranslated region (UTR in any of the three potential frames and human codon optimality (Frame 1: $R = -0.01$, $P = 0.96$; Frame 2: $R = -0.13$, $P = 0.36$; Frame 3: $R = -0.12$, $P = 0.41$) (Fig. EV1C), suggesting that this preference for nonoptimal codons is exclusive to the coding region of DENV2. Altogether, the results suggest that DENV's preference for several nonoptimal codons is unrelated with the CpG bias and therefore influenced by other driving forces.

To investigate whether this preference exists in the genomes of DENV natural isolates, the RSCU fold change was calculated from 5366 worldwide isolated DENV sequences spanning the four serotypes. The DENV isolates' relative preference for arginine codons is not equally distributed ($P < 2e-16$) (Fig. 1E). The most nonoptimal codon encoding arginine, AGA, is the most preferred codon by all four serotypes (RSCU FC: $1.32 \pm 0.07$) (Fig. 1E). The strong preference for AGA might suggest an evolutionary pressure to select this nonoptimal codon in the genome of DENV isolates. Moreover, each of the serotypes exhibits a preference for the most or second most nonoptimal codon for several other amino acids: isoleucine, glutamine, threonine, arginine, lysine, proline, glycine,

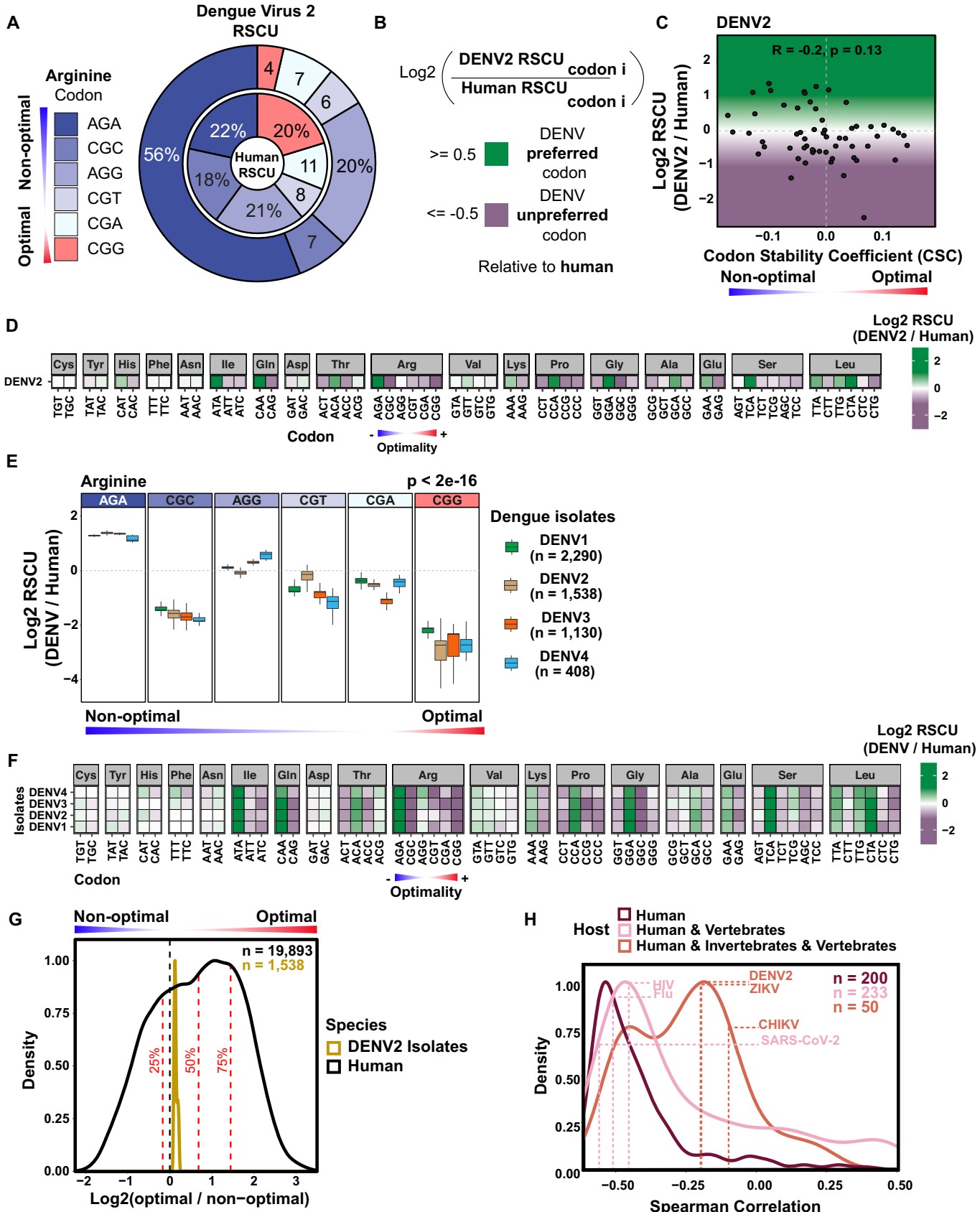

**Figure 1. Several RNA viruses preferentially use nonoptimal codons relative to humans.**

(**A**) Donut chart showing the percentage of dengue virus (DENV) Relative Synonymous Codon Usage (RSCU) and human RSCU for arginine codons. Codons are labeled based on human optimality (red = optimal, blue = nonoptimal). (**B**) The RSCU fold change was used as a metric for DENV's codon preference relative to human and it was calculated as the log2 ratio of RSCU observed in DENV compared to human per synonymous codon per amino acid. Codons showing a RSCU fold change greater than 0.5 are DENV preferred codons (green), codons with a RSCU fold change less than −0.5 are DENV unpreferred codons (purple). (**C**) Scatterplot showing the RSCU fold change (relative to human) for DENV2 and human codon stability coefficient (CSC). $R = -0.2$, $P = 0.13$, Spearman rank correlation. (**D**) Heatmap showing the RSCU fold change (relative to human) for DENV2. Codons are ordered in increasing human optimality within each amino acid. (**E**) Boxplot showing the RSCU fold change (relative to human) of the synonymous codons encoding arginine for 5366 DENV isolates spanning the four serotypes (DENV1−4). Human CSC indicated by color of codon (red = optimal, blue = nonoptimal). $P < 2e-16$, ANOVA, RSCU fold change relative to human ~ codon. For each boxplot, the median values are depicted as the center (50th percentile). The minima are the smallest data point within 1.5 times the interquartile range below the first quartile (Q1). The maxima are the largest data point within 1.5 times the interquartile range above the third quartile (Q3). The box is defined by the first quartile (Q1–25th percentile) and the third quartile (Q3–75th percentile). The difference between Q3 and Q1 represents the interquartile range. Whiskers extend from the edges of the box to the smallest (minima) and largest (maxima) values within 1.5 times the interquartile range from Q1 and Q3. (**F**) Heatmap showing the RSCU fold change (relative to human) for 5,366 DENV isolates spanning the four DENV serotypes (DENV1-4). Codons are ordered in increasing human optimality within each amino acid. (**G**) Density plot showing log2 ratio of optimal to nonoptimal codon frequency for human endogenous genes and DENV2 isolates. Quantiles for human density indicated by red dashed lines. (**H**) Density plot showing Spearman rank correlation between RSCU fold change (relative to human) and human CSC for human-infecting viruses. Viruses are split based on their host. Spearman rank correlation of particular viruses are indicated with labeled dashed lines.

glutamic acid and serine (Fig. 1F). These results indicate that this trend in preference for nonoptimal codons is conserved across DENV serotypes, despite the fact the serotypes only share ~70–75% of their amino acid and nucleotide sequences (Fig. EV1D). DENV2 isolates exhibit a proportion of optimal/nonoptimal codons below the ~34% of the human endogenous genes, indicating that natural circulating strains of DENV2 viruses preferentially use nonoptimal codons relative to humans (Fig. 1G).

To address whether the preference for nonoptimal codons was a general trend across viruses and not a specific observation for DENV, we calculated the correlation between the RSCU fold change and human CSC for 483 viruses that infect humans. The vast majority exhibit a similar preference for nonoptimal codons (negative correlation), including influenza virus, HIV and SARS-CoV-2 (Fig. 1H). Viruses that only infect humans or humans and vertebrates, presented a stronger negative correlation than viruses that infect humans and invertebrates, such as mosquito-borne viruses, suggesting that host range poses an evolutionary pressure that may shape the codon preference.

## Mosquito cells exhibit codon optimality

To investigate whether the codons preferentially used by DENV were optimal or nonoptimal in mosquito, we first determined whether codon optimality exists in *Ae. albopictus* C6/36 cells, using methods we and others have applied in other systems (Bazzini et al, 2016; Burow et al, 2018; Forrest et al, 2020; Narula et al, 2019; Presnyak et al, 2015; Wu et al, 2019). After identifying the codon optimality properties in mosquito (optimal or nonoptimal) (Fig. 2A–D), we investigated the relationship between the viral codon preference and codon optimality in each host (human and mosquito) (Fig. 3A–F).

To determine the potential regulatory properties of the codons affecting mRNA stability and protein expression (optimal or nonoptimal codons) in mosquitoes, *Ae. albopictus* C6/36 cells were treated with DMSO or with one of three different transcription inhibitors: Flavopiridol, Triptolide, and 5,6-dichloro-1-beta-D-ribofuranosylbenzimidazole (DRB) (Fig. 2A). RNA-Seq was performed at 6 h post treatment. The fold change of the mRNA level between transcription inhibitor and DMSO treatments was calculated as a metric for mRNA stability. For each codon, the

codon stability coefficient (CSC) was calculated as the Pearson correlation between mRNA stability and codon occurrence (Presnyak et al, 2015). Codons exhibiting a positive correlation are referred to as optimal codons, while codons exhibiting a negative correlation are referred to as nonoptimal codons (Fig. 2A; Dataset EV1).

The CSCs calculated using the three transcription inhibitors showed significant correlation (DRB-Flavopiridol: $R = 0.993$, $P = 3.886e-58$; DRB-Triptolide: $R = 0.966$, $P = 6.803e-37$; Triptolide-Flavopiridol: $R = 0.952$, $P = 2.606e-58$), indicating reproducibility independent of the transcription inhibitor used (Figs. 2B and EV2A). Hence, the codon optimality code of *Ae. albopictus* C6/36 cells was determined based on the combined results for the three independent transcription inhibitors used in the assays (Fig. 2B). The mosquito codon optimality code showed stronger correlation with Drosophila (Burow et al, 2018) ($R = 0.788$, $P = 4.13e-14$), than zebrafish (Bazzini et al, 2016 ($R = 0.503$, $P = 3.25e-05$) or human (Wu et al, 2019) ($R = 0.477$, $P = 8.91e-05$) (Fig. EV2B). While there are certain amino acids, such as valine, for which the codon optimality in mosquito and human are similar, there are other amino acids with different codon optimality properties. For example, CGG is the most optimal codon encoding arginine in humans, but it is the most nonoptimal in mosquito. Conversely, CGT is the most optimal codon encoding arginine in mosquito, but it is nonoptimal in human. AGA is nonoptimal in both species (Fig. 2B).

To validate that the regulatory information is encrypted in a codon-dependent manner and not simply in the nucleotide composition, we used a pair of reporters that differed by a single-nucleotide insertion (1nt frameshift). The extra nucleotide causes a frameshift that converts a "nonoptimal" sequence (enriched in nonoptimal codons) into an 'optimal' coding sequence (enriched in optimal codons), otherwise keeping the nucleotide composition nearly identical (Fig. 2C). The coding sequence of green fluorescent protein (GFP) was followed by a 1nt frameshift ribosome skipping sequence (P2A) (de Felipe et al, 2006; Donnelly et al, 2001) and a coding region enriched in either optimal or nonoptimal codons (due to 1 nucleotide frameshift) (Fig. 2C). The reporter enriched in optimal codons showed higher level of GFP compared to the counterpart reporter (enriched in nonoptimal codons) in mosquito cells co-transfected with mCherry as an internal control

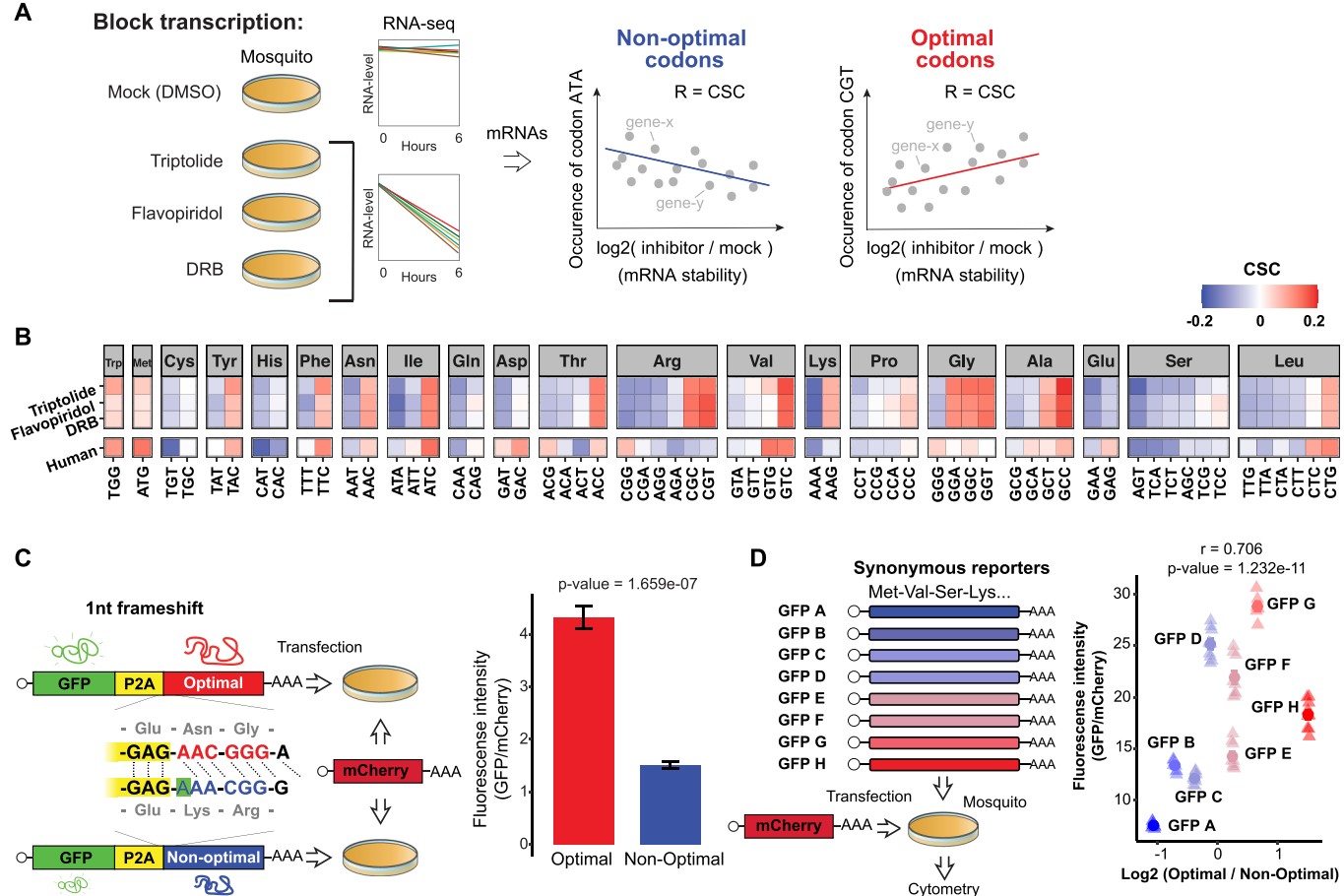

**Figure 2. Mosquito C6/36 cells exhibit codon optimality.**

(A) Diagram illustrating endogenous mRNA decay profiles approach to determine the codon optimality code in mosquito C6/36 cells. RNA-seq was performed at 6 h after mock-treated (DMSO) or blocking transcription with Flavopiridol, Triptolide, or 5,6-dichloro-1-beta-D-ribofuranosylbenzimidazole (DRB). The codon stabilization coefficient (CSC) was calculated as the Pearson correlation coefficient between the occurrence of each codon and the log2 ratio of mRNA level between treated and mock cells (metric for mRNA stability). (B) Heatmap showing the CSC calculated in human cells and in mosquito cells using the indicated transcription inhibitors to measure mRNA stability. (C) Schematic of the 1nt frameshift reporters: two mRNAs that differ in codon composition due to a single-nucleotide insertion ((A) in blue, highlighted in green), causing a frameshift. The encoding GFP fluorescent protein is followed by a cis-acting hydrolase element (P2A) and then by the coding region enriched in optimal or nonoptimal codons due to the frameshift. P2A causes ribosome skipping, thus the GFP is not fused to the optimal or nonoptimal encoded proteins. These reporters were co-transfected into *Ae. albopictus* C6/36 cells with a vector encoding for mCherry as an internal control. Barplot showing that the 1nt frameshift reporter enriched in optimal codons displayed higher GFP/mCherry fluorescent intensity than its nonoptimal counterpart measured by flow cytometry analysis. Results are shown as the averages of GFP/mCherry fluorescence intensity ± standard error of the mean from two independent experiments with five biological replicates per experiment (*P* = 1.659e−7, unpaired *t* test). (D) Illustration of 8 GFP encoding mRNAs differing only in synonymous mutations. All GFP variants were co-transfected into *Ae. albopictus* C6/36 cells with a vector encoding for mCherry as an internal control. Scatterplot showing a positive correlation between the log2 ratio of optimal to nonoptimal codon frequency for all these GFP variants (based on mosquito optimality) and GFP fluorescence intensity in mosquito C6/36 transfected cells measured by flow cytometry analysis. Results are shown as the GFP/mCherry fluorescence intensity from at least two independent experiments with three biological replicates per experiment. *R* = 0.706, *P* = 1.232e−11, Spearman rank correlation.

(*P* = 1.659e−7, unpaired *t* test) (Fig. 2C). While in mosquito cells the pair of reporters showed 2.9-fold difference in fluorescence intensity (Fig. 2C), the same reporters displayed 2.2-fold difference in human cells (Fig. EV2C), supporting the idea that several codons show distinct regulatory properties in human and mosquito cells. Moreover, as the 1nt-out of frame reporters encode different amino acids, we also compared the expression of GFP reporters which differ only in synonymous codons (Diez et al, 2022) (Fig. 2D). We observed a positive correlation between the percentage of optimal and nonoptimal codons (defined in mosquito) and the fluorescence intensity in mosquito cells transfected with each of the GFP

variants (*R* = 0.706, *P* = 1.232e−11, Spearman correlation) (Fig. 2D). All together these results indicate that the codons contain regulatory properties and confirm that codon optimality exists in mosquito cells.

## Dengue virus preferentially uses codons that are not optimal in human or mosquitoes

After defining the codon optimality code of *Ae. albopictus*, we investigated the relationship between synonymous codon preferences of DENV and codon optimality in mosquitoes. First, to

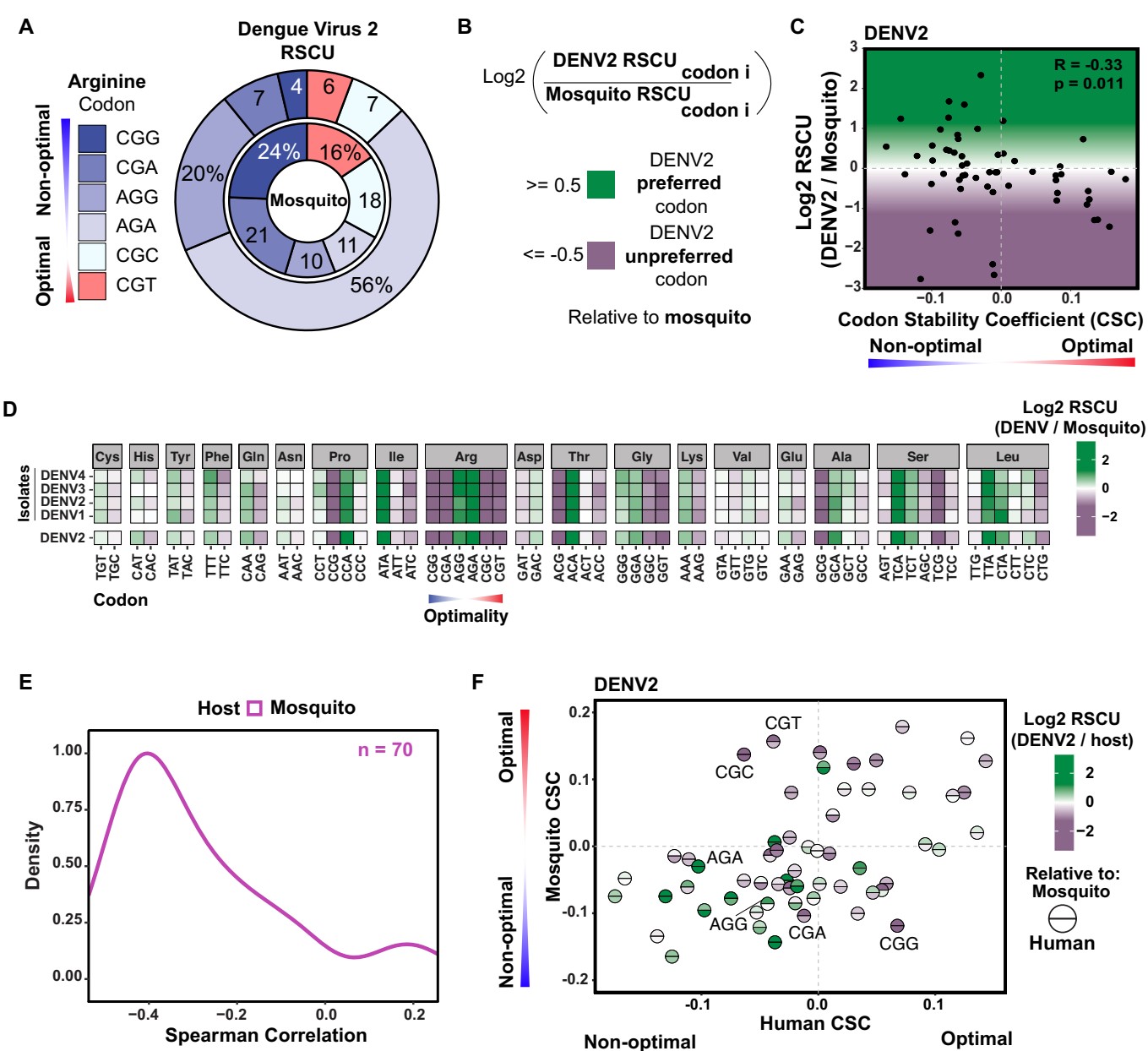

**Figure 3. Dengue virus preferentially uses nonoptimal codons relative to mosquitoes.**

(A) Donut chart showing the percentage of dengue virus (DENV) Relative Synonymous Codon Usage (RSCU) and mosquito RSCU for arginine codons. Codons are labeled based on mosquito optimality (red = optimal, blue = nonoptimal). (B) The RSCU fold change was used as a metric for DENV's codon preference relative to mosquito and it was calculated as the log2 ratio of RSCU observed in DENV compared to mosquito per synonymous codon per amino acid. Codons showing a RSCU fold change greater than 0.5 are DENV preferred codons (green), codons with a RSCU fold change less than −0.5 are DENV unpreferred codons (purple). (C) Scatterplot showing the RSCU fold change (relative to mosquito) for DENV2 strain 16681 and mosquito codon stability coefficient (CSC). $R = -0.33$, $P = 0.011$, Spearman rank correlation. (D) Heatmap showing the RSCU fold change (relative to mosquito) for 5366 DENV isolates spanning the four DENV serotypes (DENV1−4) and DENV2 strain 16681. Codons are ordered in increasing mosquito optimality within each amino acid. (E) Density plot showing Spearman rank correlation between RSCU fold change (relative to mosquito) and mosquito CSC for mosquito-specific viruses. (F) Scatterplot showing mosquito and human CSC. Each circle represents a codon. Color of the circle indicates DENV2's RSCU fold change relative to mosquito (top half) or human (bottom half). Preferred codons in both hosts (top and bottom green) cluster in the bottom left quadrant (nonoptimal in both hosts). $R = 0.46$, $P = 0.00032$, Spearman rank correlation.

determine which codons DENV preferentially uses relative to mosquito, the RSCU fold change was calculated as the ratio of RSCU observed in DENV2 to that of mosquito per amino acid (Fig. 3A,B; Dataset EV1). A significant negative correlation was observed between the RSCU fold change of DENV2 relative to

mosquito and mosquito codon optimality ($R = -0.33$, $P = 0.011$) (Fig. 3C). Specifically, for amino acids such as isoleucine, threonine, glycine, alanine, serine and leucine, the most preferred codon used by the four DENV serotypes (shown in green) relative to mosquito tends to be nonoptimal (Fig. 3D). In the case of arginine, the codon

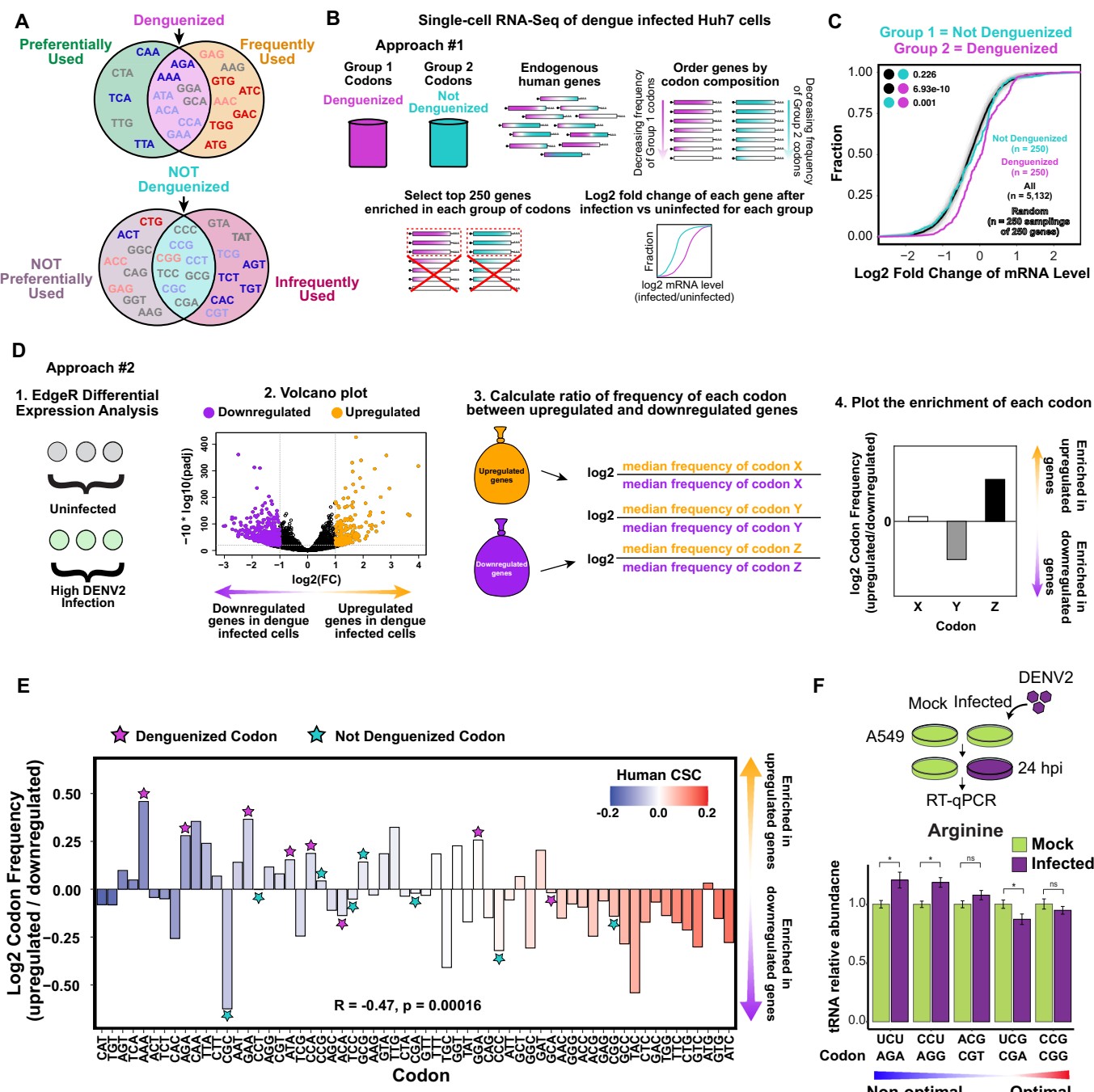

AGA is less nonoptimal in mosquitoes than in humans, but it is still nonoptimal (Fig. 2B) and strongly preferred by all DENV serotypes relative to mosquito (Fig. 3D). Moreover, the thousands of worldwide DENV isolates showed the preference for AGA relative to mosquito (Figs. 3D and EV3A). Overall, our findings suggest that DENV prefers to use codons that are nonoptimal in mosquitoes, consistent with our results in humans (Fig. 1).

Moreover, since the CpG dinucleotide bias is not present in mosquitoes (Simmonds et al, 2013), we hypothesized that any codon bias observed in viruses which only infect mosquito (and not vertebrates) would be unrelated to the CpG dinucleotide bias.

Interestingly, the vast majority of viruses which only infect mosquito displayed a negative correlation between the RSCU fold change and the mosquito CSC (Fig. 3E), suggesting that even in the absence of the host's CpG dinucleotide bias, viruses preferentially use nonoptimal codons.

To further explore the relationship between relative codon preference and host optimality, the codon optimality of mosquitoes was plotted against the codon optimality of humans (Fig. 3F). The top half of each circle (codon) was color-coded based on the RSCU fold change of DENV2 relative to mosquito and the bottom half relative to human. Codons preferentially used by DENV relative to

◄ **Figure 4. Codon-mediated regulation of gene expression in dengue-infected human cells.**

(A) Venn diagrams showing the codons that are preferentially or not preferentially used by dengue virus (DENV) relative to human, and the ones that are frequently or infrequently used in DENV2's genome. Codons that are preferentially and frequently used by DENV were called "denguenized" and the codons not preferentially and infrequently used were called "not denguenized". Color of the codon indicates human CSC (red = optimal, blue = nonoptimal, gray = neutral). (B) Diagram of the mRNA level analysis of genes enriched in "denguenized" and "not denguenized" codons (Approach #1). Human endogenous genes enriched in each codon group were identified and their mRNA level fold change upon DENV2 infection was calculated from single-cell sequencing of infected Huh7 cells using edgeR package. (C) Cumulative distribution of mRNA level fold change from human endogenous mRNAs enriched in "denguenized" and "not denguenized" codons, upon DENV2 infection. All other genes are shown in black. In total, 250 samplings of 250 random endogenous genes from "All" group shown in light gray. P values indicated, Wilcoxon rank-sum test. (D) Schematic of the codon frequency analysis of genes upregulated/downregulated upon DENV2 infection of Huh7 cells (Approach #2). Cells were ordered in increasing DENV expression and differential expression analysis was performed on uninfected and high-infection groups using edgeR package. Codon frequency of upregulated and downregulated genes was calculated and plotted as the log2 ratio of the median frequency of each codon in the group of upregulated and downregulated genes. (E) Barplot showing enrichment of each codon in the upregulated/downregulated genes. Codons with a value greater than 0 are enriched in the upregulated group, codons with a value less than 0 are enriched in the downregulated group. Codons are ordered in increasing human CSC (red bars = optimal, blue bars = nonoptimal, white bars = neutral). $R = -0.47$, $P = 0.00016$, Spearman rank correlation. Stars indicate "denguenized" (purple) and "not denguenized" (teal) codons. Nonoptimal, "denguenized" codons are enriched in upregulated genes ($P = 0.000136$, unpaired Wilcoxon test). (F) qRT-PCR analysis showing the relative quantification of arginine tRNA levels in mock -nfected and DENV infected human A549 cells. Results are shown as the averages of tRNA abundance relative to tRNAHisGTG ± standard error of the mean from two independent experiments with three biological replicates per experiment. $P = 0.0276$ for UCU, $P = 0.00469$ for CCU, $P = 0.147$ for ACG, $P = 0.0497$ for UCG, $P = 0.349$ for CCG, comparing mock and infected cells. *$P < 0.05$, unpaired $t$ test.

both human and mosquito (green top and green bottom) cluster in the lower-left quadrant. This indicates that DENV2 prefers codons that are nonoptimal in both hosts (Fig. 3F), while codons that are optimal in one or both hosts tend to not be preferentially used by DENV. In sum, DENV does not preferentially use codons that are defined as optimal in either of its hosts (human and/or mosquito).

## Human genes enriched in the codons used by dengue virus are upregulated upon dengue virus infection

Our results indicate DENV prefers nonoptimal codons relative to humans and mosquitoes. To investigate whether changes in host gene expression upon DENV infection are regulated in a codon-dependent manner, we analyzed single-cell sequencing data of DENV2 infected human Huh7 cells (Zanini et al, 2018). First, we defined a set of "denguenized" codons as the codons that are both preferentially used by DENV relative to humans (more than 0.5 log2 RSCU fold change) and are one of the 16 most frequently used codons by DENV. Conversely, we called "not-denguenized" codons as a group of codons that are not preferentially used by DENV relative to human (less than −0.5 log2 RSCU fold change) and are one of the 16 least frequently used codons by DENV (Figs. 4A and EV4A,B).

Two independent approaches were taken to investigate whether there is a codon-mediated effect on host gene expression upon DENV infection. First, two groups of genes were created by selecting the 250 human endogenous genes most enriched in "denguenized" or "not-denguenized" codons. The mRNA-level fold change between infected and uninfected cells was then calculated for each group of genes (Fig. 4B). Genes enriched in "denguenized" codons were upregulated upon infection compared to genes enriched in "not-denguenized" codons ($P = 1e-3$, Wilcoxon rank-sum test) (Fig. 4C).

In the second approach, the differentially expressed genes (up- and downregulated) between infected vs. uninfected cells were selected (adjusted $P$ value < 0.01, and fold change >1 or < −1, respectively) (Dataset EV2), and then the codon composition of each group was compared (Fig. 4D). The group of upregulated genes during DENV infection was enriched in nonoptimal codons and depleted in optimal codons, relative to the downregulated

genes ($R = -0.47$, $P = 0.00016$, Spearman correlation) (Fig. 4E). The group of upregulated genes during infection also showed a higher enrichment of the "denguenized" codons relative to the group of downregulated genes ($P = 0.000136$, unpaired Wilcoxon test) (Fig. 4E).

As tRNA level correlates with the regulatory properties of the codons (optimal or nonoptimal) (Bazzini et al, 2016; Presnyak et al, 2015; Rak et al, 2018; Wu and Bazzini, 2023; Wu et al, 2019), we hypothesized that these codon-dependent changes in gene expression may be explained by changes in the tRNA pool upon DENV infection. Interestingly, the relative level of arginine tRNA decoding AGA (nonoptimal, "denguenized") and AGG (nonoptimal) were upregulated in A549 human cells infected with DENV compared to mock-infected cells (Fig. 4F, $P < 0.05$, unpaired $t$ test). In contrast, the relative level of arginine tRNA decoding CGA (neutral, "not-denguenized") was downregulated in infected cells compared to mock-infected cells (Fig. 4F, $P < 0.05$, unpaired $t$ test). Non-significant changes in the level of arginine tRNA decoding CGT (slightly nonoptimal) and CGG (optimal) codons were observed between the infected and uninfected cells (Fig. 4F, $P > 0.05$, unpaired $t$ test). Together these results support the idea that DENV infection alters tRNA availability and may underlie the codon-based changes observed in human gene expression during infection.

## Dengue virus-synonymous mutations towards dengue virus' preferred codons tend to increase the viral relative fitness during adaptation to human or mosquito

Following the rationale that synonymous mutations are not silent from a regulatory point of view, and that DENV preferentially uses nonoptimal codons, we investigated the adaptive effect of synonymous mutations by analyzing sequencing data of DENV in in vitro directed evolution through serial passaging in a single-host cell line (Dolan et al, 2021). Analyses were focused on synonymous mutations arising after nine serial passages following the initial transfection of DENV2 strain 16681 viral RNA into human Huh7 or *Aedes albopictus* C6/36 mosquito cells (Fig. 5A). The relative fitness associated with all possible mutations (beneficial, deleterious, neutral, or lethal) across the DENV genome was calculated

 

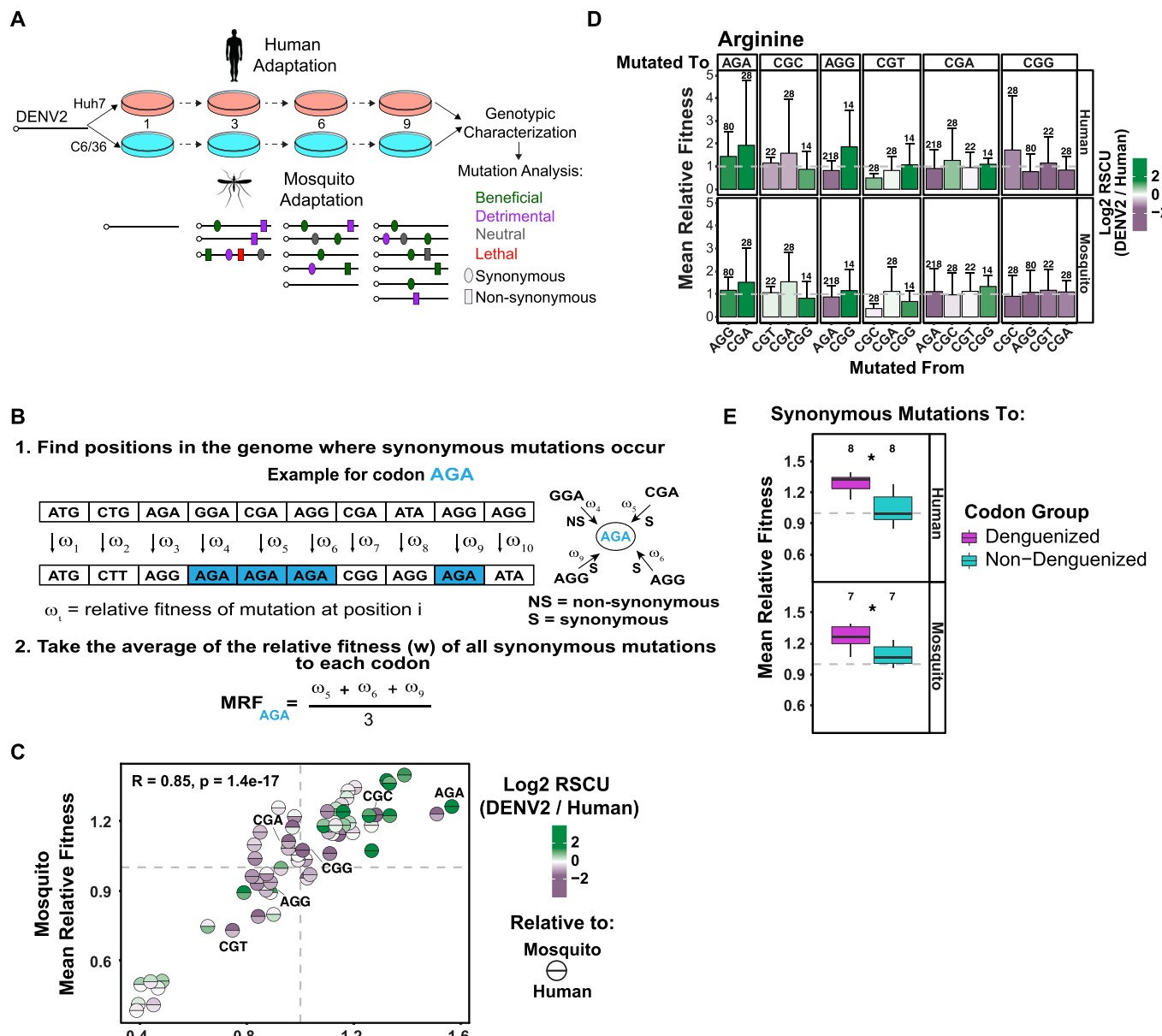

based on the frequency trajectory of a given allele over time relative to its mutation rate (Dolan et al, 2021). To evaluate the fitness effect of synonymous substitutions, we defined the Mean Relative Fitness (MRF) for each codon as the average across the relative fitness of every possible synonymous mutation to the codon across the DENV genome (Fig. 5B; Dataset EV3). The MRF of a codon represents the average relative fitness associated with a synonymous mutation to the codon, independent of the position in the genome where the mutation occurs. Therefore, an MRF of 1 indicates a neutral mutation, on average. A mutation with an MRF greater than 1 is considered beneficial on average, indicating that a synonymous mutation towards the particular codon increases viral fitness. Conversely, an MRF less than 1 suggests a deleterious mutation on average which decreases viral fitness.

Interestingly, the MRF in mosquito and human are correlated ($R = 0.85$, $P = 1.4\text{e}{-}17$, Spearman correlation) and the codons preferentially used by DENV relative to both mosquito and human (green top and green bottom) tend to have positive MRF (Fig. 5C). For example, within the codons encoding arginine, synonymous mutations going from AGG or CGA to AGA showed MRF greater than 1, while synonymous mutations from AGA to AGG or CGA showed MRF less than 1, suggesting that it is beneficial for the virus to select AGA and detrimental to lose AGA (Fig. 5D). Similar results were obtained by analyzing all the "denguenized" and "not denguenized" codons (Fig. 5E). Specifically, synonymous mutations towards "denguenized" codons showed higher MRF than mutations towards "non-denguenized" codons, suggesting that the former are more beneficial for viral fitness compared to the latter ($P = 0.021$

**Figure 5.   Relative fitness of dengue virus mutations upon infection correlates with dengue virus' codon preference relative to mosquitoes and humans.**

(A) Outline of in vitro dengue virus (DENV) evolution experiment performed by Dolan et al, DENV RNA (Serotype 2/16881/Thailand/1985) was electroporated into mosquito (C6/36) or human (Huh7) cell lines, and the resulting viral stocks were passaged for nine passages in biological duplicates. After passage, samples of the virus from each passaged population were subject to genotypic characterization by ultra-deep sequencing using the CirSeq procedure. (B) Diagram of calculation of the Mean Relative Fitness (MRF) of synonymous substitutions in the DENV genome during adaptation to human or mosquito cells. First, for each codon, we selected all synonymous mutations *to* that codon. Second, we used sequencing data from the 9th passage to identify the positions in the DENV genome where these mutations occurred and obtain the relative fitness (w) associated with each synonymous substitution at each position. Finally, the MRF of each codon was calculated by taking the average across the relative fitnesses of every possible synonymous mutation to the codon across the DENV genome. The MRF represents the average relative fitness associated with synonymous mutations to a codon independent of the position in the genome where the mutation occurred. (C) Scatterplot showing the Mean Relative Fitness (MRF) after adaptation to mosquito or human cells. Each codon, represented by a circle, was labeled with DENV2's codon preference (RSCU fold change) relative to mosquito (top half) and human (bottom half) hosts. An MRF greater than 1.0 indicates the mutation towards that particular codon increases the viral fitness, on average. DENV's preferred codons relative to both hosts (top and bottom green) cluster in the top right quadrant, suggesting synonymous mutations towards these codons increase the fitness of DENV during adaptation to mosquito and human cells. $R = 0.85$, $P = 1.4e{-}17$. Spearman rank correlation. (D) Barplot comparing the Mean Relative Fitness (MRF) of synonymous mutations within arginine during adaptation to human (top) or mosquito (bottom) cells. Bars are labeled with DENV2's codon preference (RSCU fold change) relative to human (top) or mosquito (bottom) hosts. An MRF greater than 1.0 indicates the mutation towards that particular codon increases the viral fitness, on average. Results are shown as the MRF + standard deviation measured in two biological replicates. The error bars represent +1 standard deviation around the MRF of the mutation. The number of synonymous mutations analyzed are indicated on top of each bar. (E) Boxplot showing the Mean Relative Fitness (MRF) of synonymous mutations to the group of "denguenized" (purple) and "not denguenized" (teal) codons during adaptation to human (top) or mosquito (bottom) cells. Synonymous mutations towards "denguenized" codons showed higher MRF than mutations towards "not denguenized" codons, suggesting that the former are more beneficial for viral fitness compared to the latter. Results are shown as the MRF measured in two biological replicates. The numbers on top of the boxplots show the size of the group of denguenized and not-denguenized codons analyzed ($n = 8$ for human, $n = 7$ for mosquito). For each boxplot, the median values are depicted as the center (50th percentile). The minima are the smallest data point within 1.5 times the interquartile range below the first quartile (Q1). The maxima are the largest data point within 1.5 times the interquartile range above the third quartile (Q3). The box is defined by the first quartile (Q1–25th percentile) and the third quartile (Q3–75th percentile). The difference between Q3 and Q1 represents the interquartile range. Whiskers extend from the edges of the box to the smallest (minima) and largest (maxima) values within 1.5 times the interquartile range from Q1 and Q3. $P = 0.021$ (human), $P = 0.038$ (mosquito), unpaired Wilcoxon test.

(human), $P = 0.038$ (mosquito), unpaired Wilcoxon test) (Fig. 5E). Our results indicate that adaptation of DENV to human or mosquito cells results in the selection of synonymous mutations towards DENV's preferred codons, which tend to be nonoptimal, suggesting that DENV has evolved a preferred codon usage away from host codon optimality.

## Discussion

In our study, we present a variety of experimental and analytical data that collectively suggest that the choice of synonymous codons has shaped nucleotide composition of virus genomes during host–pathogen co-evolution. Several general conclusions arise from this work. First, DENV preferentially uses nonoptimal codons and avoids codons that are defined as optimal in either human or mosquito cells (Figs. 3 and 6). Second, similar codon preference was observed in the four DENV serotypes, including thousands of worldwide isolates, despite the fact they share less than 75% amino acid similarity (Fig. 1). These points are clearly illustrated by codons encoding arginine, as AGA, the most nonoptimal codon encoding this amino acid in human, was not only the most preferentially used by DENV, but also shows the lowest dispersion of RSCU within the thousands of worldwide isolates suggesting an evolutionary pressure to select this codon (Fig. 1). Phylogenetic analyses have indicated that all four epidemic/endemic DENV serotypes evolved independently from sylvatic progenitors that utilize nonhuman primate hosts and *Aedes* vectors (Holmes and Twiddy, 2003; Wang et al, 2000), raising the possibility that selection for nonoptimal codons could be attributed to convergent evolution. Third, host (human) genes enriched in the codons preferentially and frequently used by DENV, defined as 'denguenized', are upregulated during DENV infection, suggesting a co-evolution between the codons used by the virus and by the host

genes induced upon infection (Figs. 4 and 6). The fact that the tRNA decoding the most nonoptimal and preferentially used codon encoding for arginine is upregulated upon infection; and, that the tRNA decoding the most optimal and not preferred codons by DENV are downregulated or not changing, suggests that the regulatory properties (optimal or nonoptimal) of the codons might change during infection (Figs. 4 and 6). A fourth general observation is that analysis of synonymous mutations after host-restricted serial passages in human or mosquito cells showed that synonymous mutations towards the preferred nonoptimal codons (e.g., AGA) increase DENV fitness during infection (Fig. 5). Our analysis of the correlation between codon optimality and codon preference showed that many viruses, including DENV, tend to use nonoptimal codons as defined by their host (Figs. 1 and 3). These correlations were stronger for viruses which only infect humans or humans and vertebrates, compared to viruses that infect humans and invertebrates, suggesting that codon preference is under evolutionary pressure and depends on the evolutionary distance of their host (Fig. 1). This preference for nonoptimal codons seems to be influenced by other driving forces different from CpG dinucleotide bias, since viruses which only infect mosquitoes show this preference despite the fact that mosquitoes lack the suppression of CpG dinucleotide observed in vertebrates (Gaunt and Digard, 2022; Simmonds et al, 2013) (Fig. 3).

Several hypotheses on virus-host interactions emerge from these observations. For example, virus gene expression is tightly regulated during infection. Viruses might avoid optimal codons because increasing viral RNA stability and translation efficiency might be detrimental, impeding host translation and thus triggering an early antiviral response. Our findings align with evidence suggesting that viruses sharing codon usage with their host may harm the host cells as elevated expression of viral genes would lead to depletion of tRNA (Chen et al, 2020). In addition, it is possible that the influence of codon composition on RNA structure, translation elongation, and protein

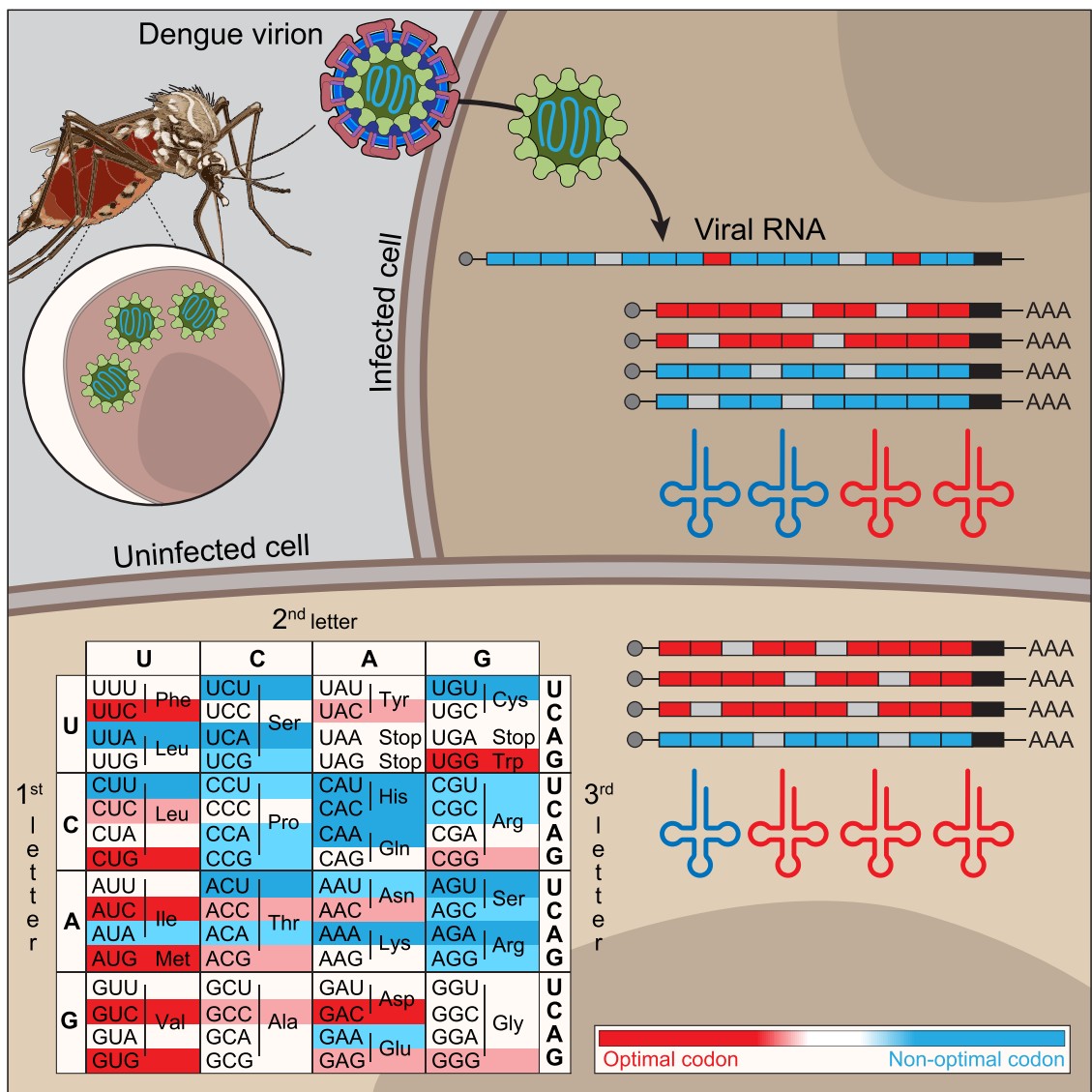

**Figure 6. Model of codon-dependent regulation during dengue virus infection.**

Dengue virus preferentially uses nonoptimal codons relative to both human and mosquito hosts. Upon infection, human genes enriched in nonoptimal codons are upregulated as well as arginine tRNA decoding the most nonoptimal and dengue preferred codon.

folding (Hanson and Coller, 2018; Liu et al, 2021; Presnyak et al, 2015; Wu et al, 2019; Yu et al, 2015) could constrain viral codon choice. Moreover, research has demonstrated that the virulence of RNA viruses can be effectively reduced by recoding viral genomes to include codon pairs underrepresented in the protein-coding sequences of their hosts. This strategy has been successfully employed in the development of attenuated vaccine candidates (Broadbent et al, 2016; Coleman et al, 2008; Gaunt et al, 2016; Le Nouen et al, 2014; Mueller et al, 2010; Shen et al, 2015; Wang et al, 2015; Yang et al, 2013). An alternative explanation proposes that RNA virus attenuation by codon pair deoptimization is driven by increases in CpG/UpA dinucleotide frequencies at the codon pair boundary, making the recoded viruses susceptible to recognition by the host innate immune response (Atkinson et al, 2014; Kunec and Osterrieder, 2016; Tulloch et al,

2014). It was shown that the host factor zinc-finger antiviral protein (ZAP) inhibits RNA virus replication in vertebrates by preferentially binding to CpG-rich RNAs and targeting them for degradation (Ficarelli et al, 2020; Ficarelli et al, 2021; Ficarelli et al, 2019; Takata et al, 2017). In this line, the least frequently used codons by DENV, such as GCG, CGA, CGC, CGT, CCG, TCG, and CGG (as shown in Fig. EV4B), are all CpG-containing codons. Therefore, we cannot rule out the possibility that viral preference for nonoptimal codons relative to their host is driven by the avoidance of CpG dinucleotide and/or specific codon pairs.

However, our research presents a set of evidence suggesting a codon-mediated effect distinct from CpG avoidance. First, DENV preferentially uses particular nonoptimal codons over other codons which do not show CpG dinucleotide in positions 1–2, 2-3 or at the

codon boundaries (Fig. 1D). Second, mosquito genes do not show signs of CpG depletion (Gaunt and Digard, 2022; Provataris et al, 2018) and underrepresentation of CpG has not been reported in insect-specific viruses (Lobo et al, 2009; Sexton and Ebel, 2019; Simmonds et al, 2013). Noteworthy, we found that mosquito and human optimality codes display differences and still viruses which exclusively infect mosquitoes also show a preference for non-optimal codons in mosquito hosts (Fig. 3E). Third, our tRNA analysis showed a correlation between changes in human tRNA levels for Arginine upon DENV infection and the codons preferentially used by DENV (Fig. 4F). Forth, while dsRNA and large dsDNA viruses do not show an underrepresentation of CpG dinucleotides in their genomes (Gaunt and Digard, 2022), our analysis reveals that viruses such as Human mastadenovirus A, Human alpha-herpesvirus 3, African swine fever virus, and Mammalian orthoreovirus 3 exhibit a preference for nonoptimal codons (Dataset EV4). Altogether, evidence indicates that both nucleotide and codon composition are under evolutionary pressures.

We hypothesize that virus infection may modulate tRNA abundance, altering host cell gene expression. Given the coevolutionary nature of host–pathogen interactions, either the virus or host cell could regulate tRNA expression upon infection. Our hypothesis suggests that the preference for nonoptimal codons offers a viral advantage, a trend evident in hundreds of human viruses (Fig. 1H). While tRNAs decoding optimal codons are highly expressed relative to their demand, tRNAs that service nonoptimal codons are normally expressed at low levels (Bazzini et al, 2016; Despic and Neugebauer, 2018; Wu and Bazzini, 2023; Wu et al, 2019). Thus, the regulation of low-abundant tRNAs may serve as a strategy to economize viral resources in the infected cell. tRNA can be regulated at multiple levels (Rak et al, 2018; Wilusz, 2015). For example, viruses such as HIV (van Weringh et al, 2011), polyomavirus (SV40) (Felton-Edkins and White, 2002), adenovirus (Gaynor et al, 1985; Hoeffler and Roeder, 1985), MHV68 (Tucker et al, 2020), Epstein Barr Virus (Felton-Edkins et al, 2006) and HSV-1 (Panning and Smiley, 1994), stimulate Pol III transcription. In addition, changes in tRNA availability in HSV-1 infected cells were linked with interferon activation (Smith et al, 2018). The changes in tRNA level have been proposed to favor translation of viral proteins (Pavon-Eternod et al, 2013; van Weringh et al, 2011). Beyond tRNA levels, tRNA modifications play a major role in tRNA function, including stability and translation fidelity (El Yacoubi et al, 2012; Suzuki, 2021; Wilusz, 2015). For instance, it has been proposed that there is a codon-specific reprogramming of translation via tRNA modification in chikungunya (CHIKV) infected human cells (Jungfleisch et al, 2022). Interestingly, AGA (Arg), GAA (Glu), AAA (Lys), CAA (Gln), and GGA (Gly) codons are preferred by both CHIKV and DENV, and they are targets of the KIAA1456 enzyme involved in the tRNA modifications that favor decoding of these A-ending codons over the G-ending ones (Jungfleisch et al, 2022). A recent preprint suggests that DENV manipulates the host tRNA epitranscriptome to promote viral replication by suppressing the host tRNA writer ALKBH (Chan et al, 2023). Moreover, tRNA-derived fragments (tRFs) can have a regulatory function (Wilusz, 2015), and, upon respiratory syncytial virus (RSV) infection, tRFs promote viral replication (Deng et al, 2015). Therefore, it is plausible that changes in tRNA levels and tRNA modifications, as well as an upregulation of tRF upon infection, might play a role in optimizing the host environment to favor translation of the viral RNA (Nunes et al, 2020).

There are other cellular contexts where tRNAs change with a concomitated change in the expression the genes enriched in those codons. For example, the tRNA repertoire varies depending on the differentiation or proliferation status of the cells and is tightly coordinated with changes in the transcriptome (Gingold et al, 2014; Wu and Bazzini, 2023). Upregulation of specific tRNAs in human breast cancer cells promotes breast cancer metastasis through a remodeling of protein expression by enhancing stability and/or ribosome occupancy of transcripts enriched for their cognate codons (Goodarzi et al, 2016). Arginine limitation in colorectal cancer cells represses arginine tRNAs, leading to ribosomal stalling at arginine codons and a proteomic shift to arginine low proteins (Hsu et al, 2023). Moreover, a higher abundance of m7G-modified tRNA Arg-TCT has been reported to drive oncogenic transformation through reshaping gene expression by enhancing stability and translation efficiency of mRNAs enriched in the corresponding AGA codon (Orellana et al, 2021). Loss of function of one central nerve-specific isodecoder of tRNA Arg-TCT induces ribosome stalling at AGA codons causing neurodegeneration in mice (Ishimura et al, 2014).

Our work raises three questions for future study: What is special about the tRNA decoding AGA, which systematically appears to be the most regulable and which is related to neurodegenerative disease, cancer, and now virus infection? Are the changes in the tRNA pool initiated upon DENV infection beneficial for the virus and/or for the host? Mechanistically, how is the tRNA availability being modulated upon viral infection? It would be interesting to know whether modulations of the tRNA pool are actively induced by the virus or by the host in response to viral infection. In summary, this work showing that DENV and other viruses preferentially use nonoptimal codons compared to their host has important implications for understanding the evolution of host–pathogen interactions.

## Methods

### Cell lines

*Aedes albopictus* C6/36 cells were obtained from ATCC [CRL-1660] and were cultured in Leibovitz's L-15 medium supplemented with 10% fetal bovine serum (FBS), 0.02% L-glutamine, 100 U/ml of penicillin, 100 μg/ml of streptomycin, and 0.25 μg/ml of amphotericin B (Fungizone) and grown at 28 °C. *Aedes albopictus* C6/36HT cells (ATCC CRL-1660), adapted to grow at 33 °C, were cultured in Leibovitz's L-15 medium supplemented with 10% fetal bovine serum (FBS), 100 U/ml of penicillin, 100 μg/ml of streptomycin, 0.3% tryptose phosphate broth, 0.02% glutamine, 1% Minimum Essential Medium (MEM) nonessential amino acid solution, and 0.25 μg/ml of amphotericin B (Fungizone). A549 cells (human lung adenocarcinoma epithelial cell line, ATCC, CCL-185) were cultured in Dulbecco's modified Eagle's medium (DMEM) supplemented with 10% FBS, 0.02% L-glutamine, 100 U/ml of penicillin, and 100 μg/ml of streptomycin. BHK-21 cells (baby hamster kidney cell line, ATCC, CCL-10) were cultured in MEM Alpha supplemented with 10% FBS, 100 U/ml of penicillin, and 100 μg/ml of streptomycin. 293T cells (human embryonic kidney

cell line, ATCC, CRL-11268) were cultured in DMEM media supplemented with 10% FBS, 0.02% L-glutamine, 100 U/ml of penicillin, and 100 µg/ml of streptomycin.

## RNA decay

*Aedes albopictus* C6/36 cells were sub-cultured in a 24-well plate at a relatively low passage and set overnight to reach 70% confluency the day of treatment. Triptolide (Sigma), Flavopiridol (Sigma) or 5,6-dichloro-1-beta-D-ribofuranosylbenzimidazole (DRB) (Sigma) were added into the well, with final concentration of 5 µM, 0.5 µM, and 30 µM, respectively, in 0.1% DMSO. Cells were directly collected in Trizol for RNA extraction at 6 h post treatment.

## RNA isolation and purification

RNA was extracted from cells using TRIzol (Invitrogen) and isopropanol precipitation according to the manufacturer's instructions. After precipitation, the RNA pellet was washed twice with ice-cold freshly prepared 75% ethanol and subsequently air-dried and resuspended in water. RNA was quantified using Qubit RNA broad range kit for RNA transfection, RNA-seq or RT-qPCR.

## Reporter design and cloning

GFP synonymous reporters (Fig. 2D) were previously designed in our laboratory (Diez et al, 2022) and renamed to reflect expected optimality in mosquito. mCherry reporter used as a transfection control (Fig. 2C,D) was designed to display neutral optimality in mosquito. The 1nt frameshift reporters (optimal and nonoptimal) were designed to have opposite codon composition (optimal and nonoptimal in mosquito) but almost identical nucleotide sequence due to a single-nucleotide insertion, causing a frameshift. All reporters were synthesized by IDT and cloned into pKM50 backbone for expression in mosquito cells. pKM50 was a gift from Kevin Maringer (Addgene plasmid # 123656; http://n2t.net/addgene:123656 ; RRID:Addgene_123656). The cloning was done by Gibson assembly with NEBuilder HiFi DNA Assembly Master Mix following the manufacturer's instructions. All sequences can be accessed in Dataset EV5.

## DNA transfection and cytometry analysis

For transfection, mosquito C6/36 cells or human 293T cells were sub-cultured in 24-well plates at a relatively low passage and set overnight to reach 70% confluency the day of transfection. Prior to transfection, all plasmids were quantified using the Qubit Fluorometric Quantification. C6/36 cells were transfected using Effectene transfection reagent based on the manufacturer's instructions. In total, 200 ng total DNA per well was added with transfection reagents. C6/36 cells were collected for cytometry analysis at 60 h post transfection. 293T cells were transfected using Lipofectamine 3000 transfection reagent based on the manufacturer's instructions. In total, 500 ng total DNA per well was added with transfection reagents. 293T cells were collected for cytometry analysis at 48 h post transfection. The fluorescence intensity of the cells was quantified in a ZE5 Cell Analyzer, using lasers and detectors for GFP (488/510) and mCherry (587/610). The cytometry data .fsc file

were analyzed using FCS Express 7, and the median intensity of the cells was used to represent fluorescence intensity. At least two independent experiments were conducted with at least three biological replicates per experiment, and the fluorescence intensity of all replicates was averaged.

## RNA transcription, transfection, and viral stock production

Plasmids containing the full-length sequences of DENV2, were linearized with XbaI and used for in vitro transcription using T7 RNA polymerase (Ambion) in the presence of m7GpppA cap analog (NEB). RNA quantification was assessed using a Qubit RNA broad range kit (Invitrogen) and RNA integrity was confirmed on 1% agarose gels. RNA transcripts were transfected into C6/36HT cells using Lipofectamine 2000 and Opti-MEM media (Invitrogen) to generate viral stocks.

## Viral titration

Viral titers were determined by plaque assays in BHK-21 cells. Cells were sub-cultured in 24-well plates, infected with serial tenfold dilutions, and covered with 1 ml of a semisolid MEM Alpha medium with 0.8% carboxymethyl cellulose (Sigma-Aldrich) and 5% FBS. After incubation at 37 °C for 6 to 7 days, 500 µl/well of formaldehyde 10% was added followed by incubation for 45 min at room temperature to fix the cells. The formaldehyde was discarded, and cells were washed five times with water and stained with crystal violet (200ul/well) for 20 min. Cells were subsequently washed 5 times with water and plaques were counted to determine the viral titer.

## Dengue virus 2 infection of human cells

Human A549 cells were sub-cultured in a six-well plate at a relatively low passage and set overnight to reach 70% confluency the day of infection. Cells were infected with DENV2 at a multiplicity of infection (MOI) ~10 in Opti-MEM medium. After adsorption at 37 °C for 1 h, infected monolayers were overlaid with DMEM supplemented with 5% FBS, 0.02% L-glutamine, 100 U/ml of penicillin, and 100 µg/ml of streptomycin. Cells were directly collected in Trizol for RNA extraction at 24 h post infection.

## qPCR of tRNA

The tRNA qPCR protocol was adapted from previously published protocols for Y-shaped adapter-ligated mature tRNA sequencing (Hsu et al, 2023; Shigematsu et al, 2017). Briefly, A549 cells were seeded into six-well plates. Twenty-four hours later, cells were mock-infected or infected with DENV2 with a multiplicity of infection (MOI) ~10. After 24 h, total RNA was extracted and then deacylated with 20 mM tris-HCl (pH 9.0) at 37 °C for 40 min. Y-shaped adapters were ligated with mature tRNA in the total RNA by T4 RNA ligase 2. The adapter sequences were as follows (capital and small letters designate DNA and RNA, respectively): Y-3′-AD, 5′-5phos/GTATCCAGTTGGAATTCTCGGGTGCCAAGG/3ddC-3′; Y-5′-AD-A, 5′-GTTCAGAGTTCTACAGTCCGACGATCACTGGATACTGga-3′; Y-5′-AD-G, 5′-GTTCAGAGTTCTACAGTCCGACGATCACTGGATACTGgg-3′; Y-5′-AD-C, 5′-GTTCAGA

GTTCTACAGTCCGACGATCACTGGATACTGgc-3′; and Y-5′-AD-U, 5′-GTTCAGAGTTCTACAGTCCGACGATCACTGGA-TACTGgu-3′. Ligation reactions were carried out with 1 µg of total RNA at 37 °C for 2 h and then 4 °C overnight. cDNA was synthesized by SuperScript IV Reverse Transcriptase (RT) (Thermo Fisher Scientific) with the common RT primer 5′-GCCTT GGCACCCGAGAATTCCA-3. The qPCR was performed with common RT primer and unique primers for different tRNAs. tRNA abundance was calculated relative to tRNA$^{His}_{GUG}$ using the $\Delta\Delta C_t$ method. Specific primers used in this study were as follows: tRNA$^{His}_{GUG}$: 5′-AGTGGTTAGTACTCTGCGTT-3′; tRNA$^{Arg}_{CCG}$: 5′-ATAAGGCGTCTGATTCCGG-3′; tRNA$^{Arg}_{UCG}$: 5′-GCCTAA TGGATAAGGCGTCTGACT-3′; tRNA$^{Arg}_{CCU}$: 5′-TGGCCT CCTAAGCCAGGGAT-3′; tRNA$^{Arg}_{ACG}$: 5′-AGTGGCGCAATG-GATAACG-3′; tRNA$_{Arg}$$^{UCU}$: 5′- GGCTCTGTGGCGCAATGGAT-3′. qPCR was done using Perfect SYBR Green FastMix Reaction Mixes, QuantaBio.

## Collection of viral genomes

### Dengue virus 2 (strain 16681) and other human viruses

The CDS of Dengue virus 2 (strain 16681) was collected from NCBI. All known viruses were collected from the RefSeq database using the links that are found within the following complete assembly file: https://ftp.ncbi.nlm.nih.gov/genomes/refseq/viral/assembly_summary.txt. Note that each link found within the assembly leads to a directory of files specific to a single virus. We downloaded the coding sequence of each virus using the fasta file containing "_cds_from_genomic.fna.gz" found within the parent directory. We then used the assembly_summary.txt file to rename the fasta files using the name of the virus. After downloading all of the viruses, we used the metadata file (https://www.ncbi.nlm.nih.gov/genomes/GenomesGroup.cgi?taxid=10239&cmd=download) to pair each virus with its host (which is present in the metadata file). This way, we were able to create a table with all of the virus names, their coding sequence, and their host. Finally, we filtered all of the viruses to collect only the ones that infect humans (Dataset EV4). Mosquito-specific viruses were manually filtered based on a previous study by (Agboli et al, 2019) (Dataset EV6).

### Dengue virus types 1−4 isolates

The genomes of dengue virus types 1−4 isolates were downloaded from Virus Pathogen Database and Analysis Resource (ViPR—https://www.bv-brc.org/) (Pickett et al, 2012). In order to download the CDS sequences of the isolates, the following filters were used in the genome search: Host Selection: Human, Mosquito, Type: Dengue virus 1, Dengue virus 2, Dengue virus 3, Dengue virus 4, Complete Genome Only: TRUE. All genomes were selected and the CDS fastas were downloaded.

### Dengue virus 2 (strain 16681) 3′ UTR sequence

The 3' UTR sequence of Dengue virus 2 (strain 16681) was collected from https://www.ncbi.nlm.nih.gov/nuccore/U87411.

### Calculation of relative synonymous codon usage (RSCU)

While *codon usage* is a frequently used metric, it is inherently biased. Each synonymous codon within an amino acid is not used equally—some are preferred more than others. Relative Synonymous Codon Usage (RSCU) was proposed to measure this codon usage bias (Sharp and Li, 1987). The RSCU of a codon *j* of amino acid *i* is calculated using the following formula:

$$RSCU_{ij} = \frac{x_{ij}}{\frac{1}{n_i}\sum_{j=1}^{n_i} x_{ij}}$$

where $x_{ij}$ denotes the frequency (number of occurrences) of codon *j* for the *i*th amino acid, and $n_i$ denotes the number of synonymous codons encoding the *i*th amino acid. The formula's output is a real value between 0 and $n_i$, inclusive. An RSCU of 0 signifies that the codon is never used and an RSCU of $n_i$ indicates that codon *j* is used exclusively for the amino acid *i*. For the purpose of calculating the RSCU fold changes, we do not divide by $1/n_i$ since the number of synonymous codons will cancel out. For ease of visualization, we represent the RSCU without the factor of $n_i$ as a percentage (as shown in Figs. 1A and 3A). Otherwise, we report RSCUs in their intended form, ranging from 0 to $n_i$, inclusive. The RSCUs of the viral genomes and species genomes is calculated by first summing the codon counts throughout the collection of sequences before applying the formula above. For each serotype of the isolated dengue sequences, the RSCUs are calculated as the median RSCUs of the isolates.

## Calculation of codon optimality in *Ae. albopictus* C6/36 cells

The codon optimality of C6/36 cells is calculated using the **T**ranscripts **P**er **M**illion of endogenous genes in the four sample treatments (DMSO, DRB, Flavopiridol, Triptolide). First, genes with less than or equal to 5 TPMs in the DMSO sample are removed. Genes that have 0 TPMs in the three transcription inhibitor treatments (DRB, Flavopiridol, Triptolide) are also removed. The stability of each gene in the transcription inhibitor treatments is calculated as the log2 of treatment TPM divided by the TPM of the DMSO treatment. For example:

$$stability = \log 2(TPM_{DRB}/TPM_{DMSO})$$

For each codon, genes are removed that contain 0 occurrences of the codon. With the remaining genes, the Pearson correlation is taken between the stability (*x* axis) and the codon frequency (*y* axis; percentage). The Pearson correlation between these two variables is defined as the **C**odon **S**tability **C**oefficient (CSC). Codons with a CSC > 0 are denoted as optimal codons, while codons with a CSC < 0 are denoted as nonoptimal codons.

## Classification of codons according to dengue usage and preference relative to humans

We split the 61 codons into groups defined by dengue usage and dengue preference relative to humans. We first calculated the codon usage of DENV2 strain 16681, defined as the proportion of each codon within the CDS of the virus (i.e., number of codon *x* / total number of codons in the CDS). We then formed two groups: "Frequently Used" (the 16 codons with the highest codon usage) and "Infrequently Used" (the 16 codons with the lowest codon usage). As described in Fig. 1A,B, we then calculated the log2 RSCU fold change of DENV2 strain 16681 relatives to human (RSCU calculation can be seen in "Methods"). From there, we defined another two groups: "Preferentially Used" (codons which have a

log2 RSCU fold change relative to humans that is greater than or equal to 0.5 (~1.41 ratio of virus to human)) and "Not Preferentially Used" codons which have a log2 RSCU fold change relative to humans that is less than or equal to −0.5 (~0.71 ratio of virus to human)). Lastly, we used the four groups above to create two final groups: "Denguenized" (codons that are both "frequently used" and "preferentially used") and "Not Denguenized" (codons that are both "infrequently used" and "not preferentially used"). A schematic of which codons fall in each group can be found in Fig. 4A.

### Calculation of FPKM from single-cell sequencing data

Rows of the raw counts file that are not the Dengue transcript or a human endogenous gene beginning with "ENSG" are removed. Counts Per Million (CPM) of the remaining 60,620 rows are calculated using the cpm function from the edgeR package in R. The cells (columns) that contain zero Dengue CPM are removed from the table and saved separately. Ventiles (20-quantiles) of Dengue CPM are calculated using the remaining cells. High-infection cells are defined as having Dengue CPM between the 80th and 90th percentiles, inclusive, resulting in 100 cells. The first 100 cells containing zero Dengue CPM are denoted as non-infected cells. The column names of these 200 cells are used to extract the cells from the raw counts file and the row for the Dengue transcript is removed. The result is a table of raw counts with 60,619 rows corresponding to human endogenous genes and 200 columns corresponding to 100 uninfected cells and 100 infected cells. Using the edgeR package, a DGEList object is created from the raw counts table. Normalization factors and dispersion are calculated using the calcNormFactors and estimateDisp functions. No filtering by expression is performed in order to retain all genes. Using the GenomicFeatures R package, a TxDb object is created from the release 102 GRCh38 Ensembl genome assembly gtf file. Gene lengths are extracted from the TxDb object and the Fragments Per Kilobase Million (FPKM) are calculated using the fpkm function in edgeR. The median FPKM for each gene in the uninfected cells and infected cells is then calculated. The result is a table of FPKM values with 60,619 rows corresponding to human endogenous genes and two columns corresponding to the median FPKM in the uninfected and infected cells.

### Gene expression analysis upon dengue virus infection

To determine whether there is a statistical difference between the enrichment of the "denguenized" codons in the upregulated group of genes vs the downregulated group of genes, we performed a Wilcoxon rank-sum test.

### Calculation of mean relative fitness of DENV2 populations

We define a metric called Mean Relative Fitness (MRF) to estimate the average fitness effect of a synonymous mutation towards each codon. In order to calculate MRF, we focus on the sequences of virus populations collected after nine serial passages following the initial transfection of DENV2 strain 16681 viral RNA into human Huh7 or *Ae. albopictus* C6/36 mosquito cells (Dolan et al, 2021). At each codon position within the dengue genome, we are provided with a relative fitness score for each observed mutation from the original codon to another codon. A breakdown of relative fitness scores follows: 0 (lethal mutation), <1 (deleterious mutation), ~ 1 (neutral mutation), >1 (beneficial mutation). To calculate the MRF of a codon, say AGA in humans, we first find all of the positions where a codon mutates *towards* AGA in both of the biological replicates. We discard positions where this mutation is not synonymous. Then, we extract the relative fitness scores of each of these synonymous mutations (call it $\omega_i$ —the relative fitness score of mutation to AGA at position $i$). This analysis is performed in the two biological replicates per cell line. Lastly, we take the average of all of these relative fitness scores and define the result to be the Mean Relative Fitness of codon AGA. This metric gives us an estimate of the average fitness effect of synonymous mutations towards AGA, regardless of position within the genome. We use the mutations from both biological replicates to calculate the Mean Relative Fitness for each species to reduce the variance of our calculation.

## Data availability

Original data underlying this manuscript can be accessed from the Stowers Original Data Repository at https://www.stowers.org/research/publications/libpb-2402. Sequencing data have been deposited in the NCBI Gene Expression Omnibus, GSE234878.

The source data of this paper are collected in the following database record: biostudies:S-SCDT-10_1038-S44320-024-00052-7.

## Peer review information

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

## Acknowledgements

The authors thank Dr Robb Krumlauf and Sara Bancroft (Stowers Institute) for suggestions and critical reading of the manuscript, and Mark Miller for designing the graphical abstract. Authors also thank the following Stowers Core facilities: Cells, Tissues and Organoids Center, Media Prep, Sequencing and Discovery Genomics, Cytometry, Automation and PCR Technology, and Computational Biology. The authors are thankful to members of the Bazzini laboratory for discussions, especially Gabriel da Silva Pescador, Anthony Treichel, and Dr Qiushuang Wu for technical support. The authors thank the Zeitlinger laboratory for providing reagents. This study was supported by the Stowers Institute for Medical Research. AAB and DEA were awarded a Pew Innovation Fund and AAB with the US National Institutes of Health (NIH-R01 GM136849 and NIH R21OD034161). This work was performed as part of thesis research for LAC, Graduate School of the Stowers Institute for Medical Research.

## Author contributions

**Luciana A Castellano**: Conceptualization; Data curation; Formal analysis; Validation; Investigation; Methodology; Writing—original draft; Writing—review and editing. **Ryan J McNamara**: Data curation; Formal analysis; Investigation; Methodology; Writing—original draft. **Horacio M Pallarés**: Methodology. **Andrea V Gamarnik**: Resources. **Diego E Alvarez**: Conceptualization; Funding acquisition; Writing—original draft; Writing—review and editing. **Ariel A Bazzini**: Conceptualization; Resources; Supervision; Funding acquisition; Writing—original draft; Project administration; Writing—review and editing.

Source data underlying figure panels in this paper may have individual authorship assigned. Where available, figure panel/source data authorship is listed in the following database record: biostudies:S-SCDT-10_1038-S44320-024-00052-7.

## Disclosure and competing interests statement

The authors declare no competing interests.

# Expanded View Figures

**Figure EV1.  Dengue virus serotypes preferentially use nonoptimal codons relative to human.**

(A) Heatmap showing human CSC. Optimal codons highlighted in red, nonoptimal codons highlighted in blue. Scale bar indicated. (B) Scatterplot showing the RSCU fold change (relative to human) for DENV1, DENV3 and DENV4 and human codon stability coefficient (CSC). $R = -0.31$, $P = 0.016$ for DENV1, $R = -0.28$, $P = 0.034$ for DENV3, $R = -0.26$, $P = 0.047$ for DENV4. Spearman rank correlation. (C) Scatterplot showing the RSCU fold change (relative to human) for DENV2 3'UTR in the three frames and human codon stability coefficient (CSC). Spearman correlation coefficients and p values indicated, Spearman rank correlation. (D) Matrix showing the four DENV serotypes' (DENV1–4) similarity. Lower triangle indicates amino acid similarity, upper triangle indicates nucleotide similarity.

▶

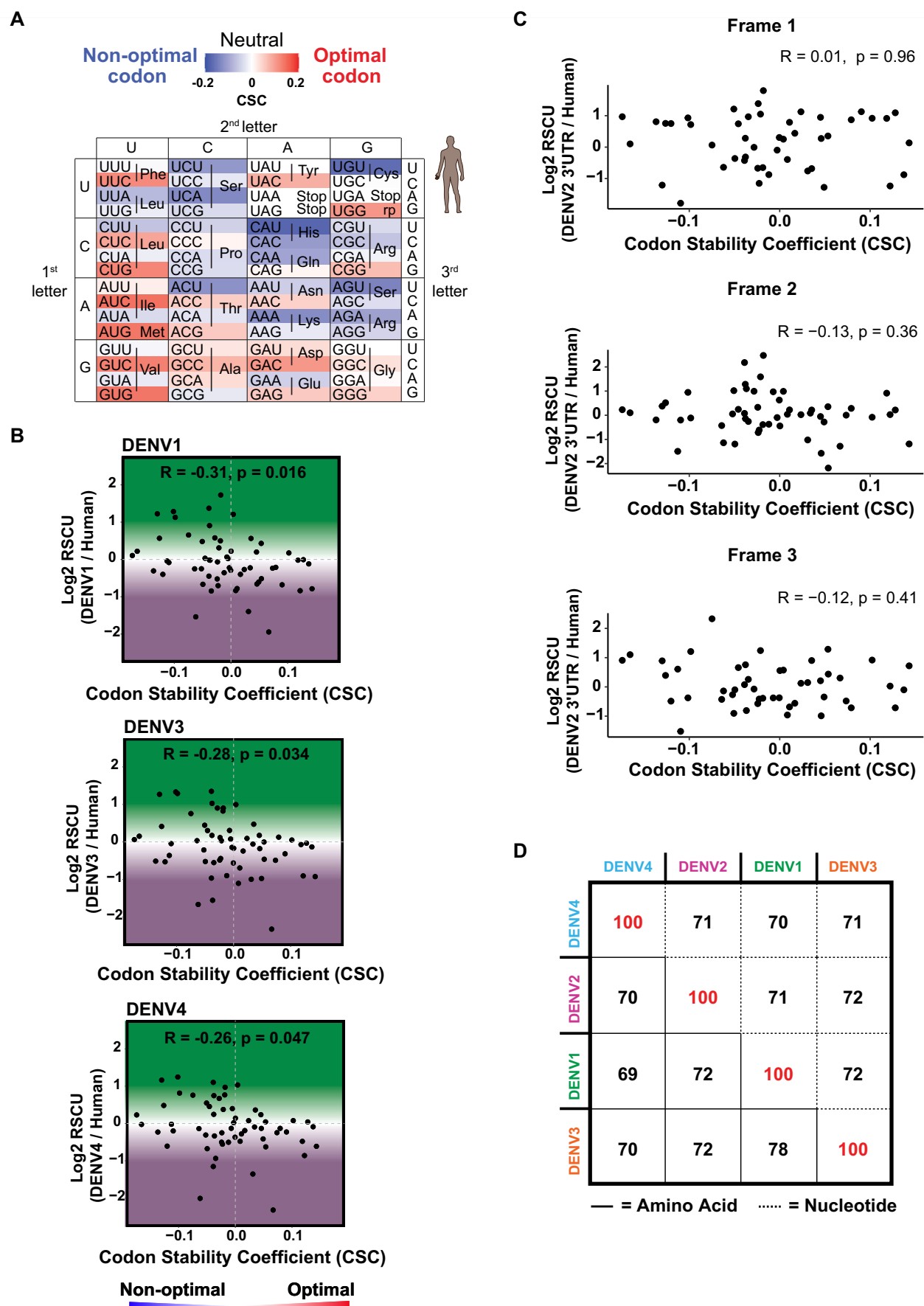

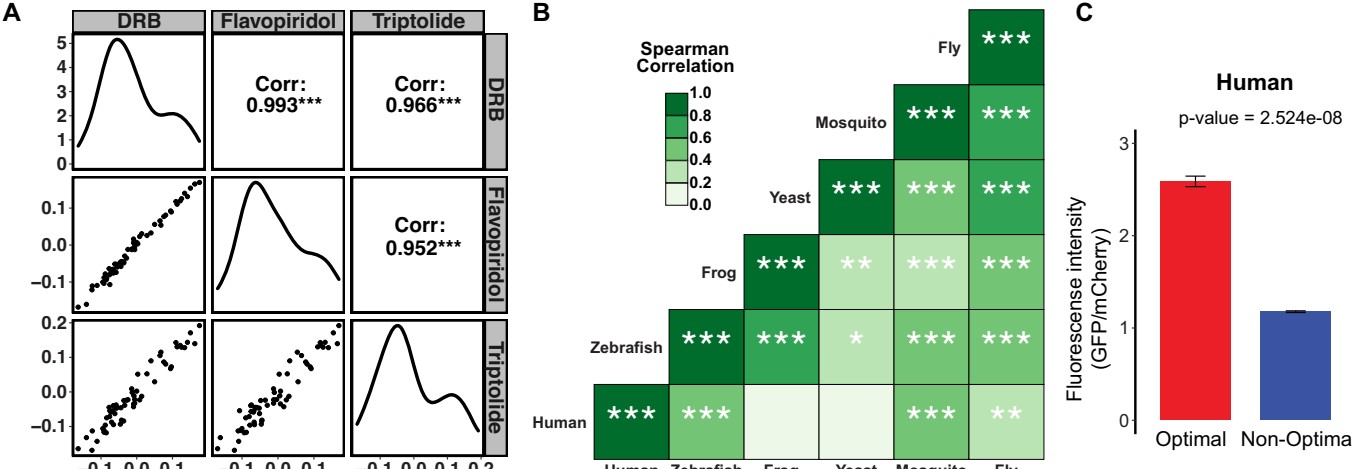

**Figure EV2. Codon optimality in mosquito and across species.**

(A) Pair plot showing the CSC calculated in mosquito C6/36 cells using indicated transcription inhibitors. Lower triangle shows scatterplots of CSCs between inhibitors. Diagonal shows density plot of CSC for each inhibitor. Upper triangle shows Pearson correlation coefficient of CSCs between inhibitors. (B) Heatmap showing Spearman rank correlations between known CSCs in indicated species. $P = 8.91e{-}05$ for Human/Mosquito, $P = 1.08e{-}05$ for Human/Zebrafish, $P = 4.13e{-}01$ for Human/Frog, $P = 0.343$ for Human/Yeast, $P = 7.94e{-}03$ for Human/Fly, $P = 3.25e{-}05$ for Mosquito/Zebrafish, $P = 9.29e{-}03$ for Mosquito/Frog, $P = 0.000181$ for Mosquito/Yeast, $P = 4.13e{-}14$ for Mosquito/Fly, $P = 8.70e{-}09$ for Zebrafish/Frog, $P = 0.0295$ for Zebrafish/Yeast, $P = 4.87e{-}06$ for Zebrafish/Fly, $P = 0.00562$ for Frog/Yeast, $P = 9.31e{-}06$ for Frog/Fly, $P = 1.97e{-}07$ for Yeast/Fly. Color of the tile indicates correlation coefficient. $*P < 0.05$, $**P < 0.01$, $***P < 0.001$. (C) 1nt frameshift reporters designed based on mosquito codon optimality were co-transfected into 293T human cells with a vector encoding for mCherry as an internal control. Bar plots showing that the 1nt frameshift reporter enriched in optimal codons displayed higher GFP/mCherry fluorescence intensity than its nonoptimal counterpart measured by flow cytometry analysis. Results are shown as the averages of GFP/mCherry fluorescence intensity ± standard error of the mean from two independent experiments with four biological replicates per experiment ($P = 2.524e{-}8$, unpaired $t$ test).

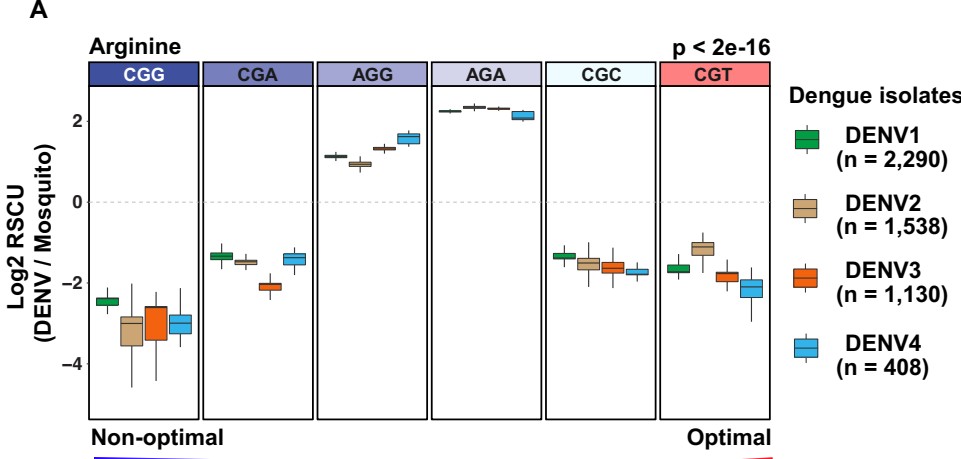

**Figure EV3.   Arginine codon preference of dengue virus isolates relative to mosquito.**

(A) Boxplot showing the RSCU fold change (relative to mosquito) of the synonymous codons encoding arginine for 5,366 DENV isolates spanning the four serotypes (DENV1−4). Mosquito CSC indicated by color of codon (red = optimal, blue = nonoptimal). *P* < 2e-16, ANOVA, RSCU fold change relative to mosquito ~ codon. For each boxplot, the median values are depicted as the center (50th percentile). The minima are the smallest data point within 1.5 times the interquartile range below the first quartile (Q1). The maxima are the largest data point within 1.5 times the interquartile range above the third quartile (Q3). The box is defined by the first quartile (Q1–25th percentile) and the third quartile (Q3–75th percentile). The difference between Q3 and Q1 represents the interquartile range. Whiskers extend from the edges of the box to the smallest (minima) and largest (maxima) values within 1.5 times the interquartile range from Q1 and Q3.

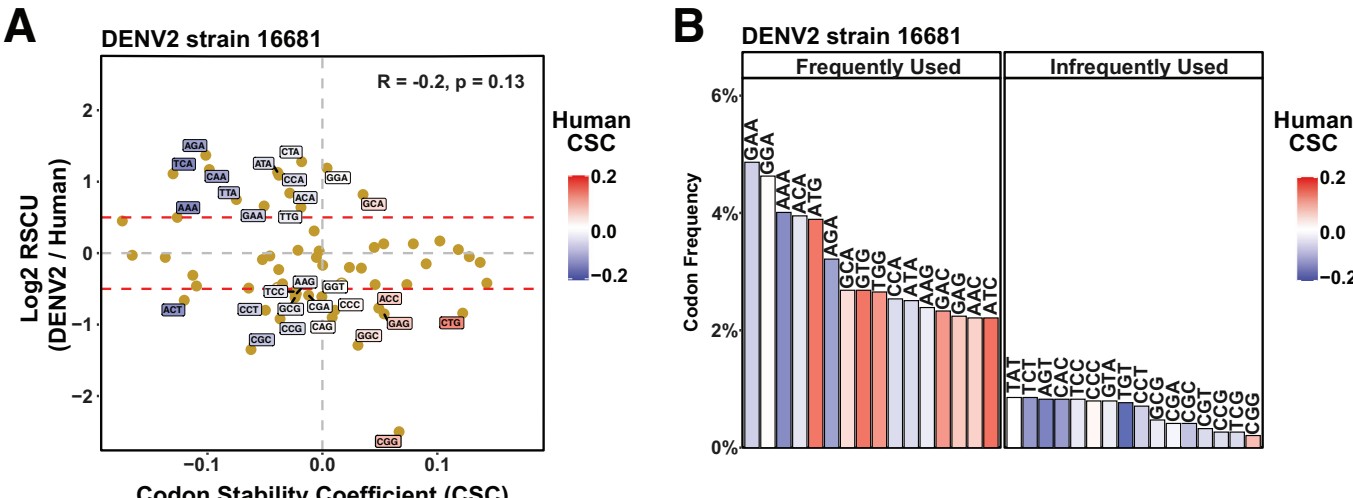

**Figure EV4. Classification of dengue virus 2 codons based on preference relative to human and usage.**

(A) Scatterplot showing the RSCU fold change (relative to human) for DENV2 strain 16681 and human codon stability coefficient (CSC). Labels indicate 'preferentially used' (log2(RSCU fold change relative to human) ≥0.5) and 'not preferentially used' (log2(RSCU fold change relative to human) ≤ −0.5). $R = -0.2$, $P = 0.13$, Spearman rank correlation. (B) Barplot showing the frequency of the 16 most used ('frequently used') and the 16 least used ('infrequently used') codons in the DENV2 strain 16681 genome.

