## [Peer Review File · Molecular Systems Biology]

Dengue virus preferentially uses human and mosquito non-optimal codons

Luciana Castellano, Ryan McNamara, Horacio Pallarés, Andrea Gamarnik, Diego Alvarez, and Ariel Bazzini

Corresponding author(s): Ariel Bazzini (arb@stowers.org)

Review Timeline:

Submission Date:	29th Jan 24
Editorial Decision:	28th Feb 24
Revision Received:	6th May 24
Editorial Decision:	3rd Jun 24
Revision Received:	25th Jun 24
Accepted:	26th Jun 24

Editors: Maria Polychronidou and Jingyi Hou

Transaction Report:

28th Feb 2024

Manuscript Number: MSB-2024-12244-T

Title: Dengue virus preferentially uses human and mosquito non-optimal codons

Dear Dr. Bazzini,

Thank you again for submitting your work to Molecular Systems Biology. We have now heard back from the three reviewers who agreed to evaluate your study. As you will see below, the reviewers find the presented findings interesting for the field. However, they raise several concerns, which we would ask you to address in a revision.

I think that the reviewers' recommendations are clear and I therefore see no need to repeat the comments listed below. Of note, reviewer #3 raises several issues related to the need to include additional controls and analyses to better support the main conclusions. All issues raised by the reviewers would need to be satisfactorily addressed. As you may already know, our editorial policy allows in principle a single round of major revision. It is therefore essential to provide responses to the reviewers' comments that are as complete as possible. If you have any questions of if you would like to discuss your revision plan with me, please feel free to get in touch.

On a more editorial level, we would ask you to address the following points:

- Please provide a .doc version of the manuscript text (including legends for main Figures and EV Figures) and individual production quality figure files for the main Figures and EV Figures (one file per figure).
- Please include 5 keywords.
- We have replaced Supplementary Information by the Expanded View (EV format). In this case (unless the number of EV figures becomes > 6 during revision), all additional figures can be provided as EV Figures. Please provide one file per EV Figure. Their legends should be included in the manuscript text. For detailed instructions regarding expanded view please refer to our Author Guidelines: .
- Supplementary Tables EV1-EV6 should be provided as EV Datasets. Please provide one file per EV Dataset. In each file, a description of the table/dataset should be provided in a separate tab.
- Please provide a "standfirst text" summarizing the study in one or two sentences (approximately 250 characters), three to four "bullet points" highlighting the main findings and a "synopsis image" (exactly 550px width and max 400px height, jpeg or png format) to highlight the paper on our homepage.
- All Materials and Methods need to be described in the main text. We would encourage you to use 'Structured Methods', our new Materials and Methods format. According to this format, the Material and Methods section should include a Reagents and Tools Table (listing key reagents, experimental models, software and relevant equipment and including their sources and relevant identifiers) followed by a Methods and Protocols section in which we encourage the authors to describe their methods using a step-by-step protocol format with bullet points, to facilitate the adoption of the methodologies across labs. More information on how to adhere to this format as well as downloadable templates (.doc or .xls) for the Reagents and Tools Table can be found in our author guidelines: . An example of a Method paper with Structured Methods can be found here:
- Please include a Data availability section describing how the data and code have been made available. This section needs to be formatted according to the example below:
The datasets and computer code produced in this study are available in the following databases:
 - Chip-Seq data: Gene Expression Omnibus GSE46748 (<https://www.ncbi.nlm.nih.gov/geo/query/acc.cgi?acc=GSE46748>)
 - Modeling computer scripts: GitHub (<https://github.com/SysBioChalmers/GECKO/releases/tag/v1.0>)
 - [data type]: [full name of the resource] [accession number/identifier] ([doi or URL or identifiers.org/DATABASE:ACCESSION])
- The link to the Stowers original data repository does not work. In any case, we would encourage you to deposit data for which no suitable public database exists in a database for unstructured data (for example: BioStudies, Dryad, Zenodo, Figshare) or provide them as EV Datasets, if the file size allows this.
- For data quantification: please specify the name of the statistical test used to generate error bars and P values, the number (n) of independent experiments (specify technical or biological replicates) underlying each data point and the test used to calculate p-values in each figure legend. The figure legends should contain a basic description of n, P and the test applied. Graphs must include a description of the bars and the error bars (s.d., s.e.m.).

- Please include a "Disclosure & Competing Interests Statement" in the main text.
 - The References should be formatted according to the Molecular Systems Biology reference style (i.e., ordered alphabetically and listing the first 10 authors followed by et al).
 - When you resubmit your manuscript, please download our CHECKLIST (<https://bit.ly/EMBOPressAuthorChecklist>) and include the completed form in your submission.
- *Please note* that the Author Checklist will be published alongside the paper as part of the transparent process (<https://www.embopress.org/page/journal/17444292/authorguide#transparentprocess>).

If you feel you can satisfactorily deal with these points and those listed by the referees, you may wish to submit a revised version of your manuscript. Please attach a covering letter giving details of the way in which you have handled each of the points raised by the referees. A revised manuscript will be once again subject to review and you probably understand that we can give you no guarantee at this stage that the eventual outcome will be favorable.

Kind regards,

Maria

Maria Polychronidou, PhD
Senior Editor
Molecular Systems Biology

We realize that it is difficult to revise to a specific deadline. In the interest of protecting the conceptual advance provided by the work, we recommend a revision within 3 months (28th May 2024). Please discuss the revision progress ahead of this time with the editor if you require more time to complete the revisions. Use the link below to submit your revision:

IMPORTANT: When you send your revision, we will require the following items:

1. a letter with a detailed description of the changes made in response to the referees. Please specify clearly the exact places in the text (pages and paragraphs) where each change has been made in response to each specific comment given
2. When assembling figures, please refer to our figure preparation guideline in order to ensure proper formatting and readability in print as well as on screen:

See also figure legend guidelines: <https://www.embopress.org/page/journal/17444292/authorguide#figureformat>

3. Please note that corresponding authors are required to supply an ORCID ID for their name upon submission of a revised manuscript (EMBO Press signed a joint statement to encourage ORCID adoption).

(<https://www.embopress.org/page/journal/17444292/authorguide#editorialprocess>)

Currently, our records indicate that the ORCID for your account is 0000-0002-2251-5174.

Link Not Available

*** PLEASE NOTE *** As part of the EMBO Press transparent editorial process initiative (see our Editorial at <https://dx.doi.org/10.1038/msb.2010.72>), Molecular Systems Biology publishes online a Review Process File with each accepted manuscripts. This file will be published in conjunction with your paper and will include the anonymous referee reports, your point-by-point response and all pertinent correspondence relating to the manuscript. If you do NOT want this File to be published, please inform the editorial office at msb@embo.org within 14 days upon receipt of the present letter.

Reviewer #1:

Codons not only define the amino acid sequence of polypeptides but also affect the amount of proteins to be synthesized through multiple mechanisms. Codon optimality is the recently unveiled effects of synonymous codon choice on mRNA stability and translational efficiency, which are deeply connected to tRNA availability. Whereas codon optimality has been reported in multiple species, from bacteria to humans, how codon optimality promotes or restricts inter-species gene expression has been poorly characterized. In this study, the authors investigated the impact of codon optimality on the infection of viruses to their hosts using mosquito-borne dengue virus (DENV) as a model. The authors first compared codon usage bias in DENV and one of its host species, humans, and showed that DENV preferentially uses codons defined as nonoptimal in humans, exemplified by the arginine AGA codon. The authors then experimentally determined codon optimality in another host, mosquitos, by measuring mRNA stability genome-wide in cultured cells and showed that DENV preferentially uses codons defined as nonoptimal in mosquitoes, too, even though codon optimality differs between the two host species. To investigate the connection between DENV codon usage bias upon infection, the authors analyzed single-cell RNA-Seq data of DENV2-infected human Huh7 cells. Interestingly, human mRNAs upregulated upon DENV2 infection were enriched with codons preferentially used in the dengue virus. The upregulation of mRNAs was accompanied by the upregulation of tRNAs, including the tRNA corresponding to the dengue-enriched AGA codon. Finally, the authors analyzed the mutation rate of the dengue virus genome during in-vitro evolution through serial passaging in human and mosquito cells. The selection of synonymous mutations occurred towards DENV's preferred codons, suggesting the advantage of viral-type codon usage bias in viral infection.

Overall, this study revealed a very interesting relationship between viral codon usage, host codon optimality, and viral infection. The results suggest that the viruses take advantage of codon optimality, at least partially, by controlling the host tRNA expression for their successful infection and propagation. This study opens a new avenue for future research regarding the mechanism of tRNA expression changes by viruses and related novel therapeutic approaches. This study will be of great interest in RNA biology, viral infection and evolution, and related medical fields. The experiments were performed elegantly, the results were analyzed appropriately and described clearly, and the discussion was dense and thoughtful. With a few minor comments listed below, I fully support the publication of this exciting paper.

Minor comments

In addition to codon optimality, several previous studies, including the one by the authors, suggested that amino acid optimality also affects mRNA stability. In this sense, I wonder if the changes in the human mRNA level upon DENV infection could be understood partly with amino acid optimality. The distinction between the codon and amino acid optimality effects could be complex. Still, the analysis of amino acid optimality might reveal the effect of viral infection on host amino acid metabolism in addition to the tRNA expression change.

Page 8, bottom, "arginine tRNA decoding CGT (slightly nonoptimal) and CCG (optimal) codons." The latter codon should be CGG (Arg), not CCG (Pro).

Reviewer #2:

In this manuscript, Castellano and colleagues report the intriguing observation that viruses appear to use non-optimal codons in their genome when compared to the codon usage of their host. The authors focus their analysis on Dengue virus (DENV) and show that its Relative Synonymous Codon Usage (RSCU) differs drastically than the human one. This seems to be the case for the Arg codon, but also more globally for a large number of other codons. Interestingly, this property is conserved in all four DENV serotypes, and the authors also found a negative correlation between RSCU fold change and human codon stability coefficient for a large collection of human as well as human and invertebrate specific viruses. Then, the authors determine the RSCU in mosquito cells, which indeed allows them to show that it does exist in this specie as well, although it differs from the human one. Strikingly, DENV RSCU also differs from the mosquito's and the virus tends to use again non optimal codons compared to the mosquito usage. Finally, the authors reanalyze previously published single-cell sequencing data as well as sequencing data of DENV following serial passaging in human or mosquito cells. This allows them to draw two important conclusions, one is that human mRNAs that are upregulated upon infection are enriched in suboptimal codons, and the other is that synonymous mutations that accumulate during passaging of the virus in one given cell type tend to favor the changes to DENV preferred codons, i.e. non optimal codons. This is a thought-provoking study that has important implications for our understanding of host-virus co-evolution, in particular to understand how the pathogen evolves to go around its host defense mechanisms. It is not completely clear though in what respect the usage of non-optimal codons benefits the virus, but the presented data appear to be solid and mostly back up the authors' conclusions. There are a few points that need to be addressed to clarify some aspects of the study and/or strengthen the authors' hypothesis.

1. While the analysis presented by the authors clearly indicates the preferred usage of sub-optimal codons by DENV, one interesting control could be to redo the same analysis by only looking at the non-coding portions of the viral genome. At the moment, the authors focused only on CDS, which perfectly makes sense, but what happens when using sequences that are not translated. The same could be done with all viruses that have been used for Figure 1H.
2. Based on the data contained in Table EV5, it seems that the list of viruses that has been used by the authors to generalize their observations contains both RNA and DNA viruses, including viruses that are known to infect their hosts for a very long time by going into latency (e.g. herpesviruses) or persistent infection (e.g. hepatitis C virus). One would expect that for these viruses,

the codon optimality should tend to align with the one of the host. The same could be true for retroviruses, which integrate into the host genome. Is this something that would be worth to check?

3. Regarding the reported that has been used in Figure 2C, what is the effect of the frameshift on mRNA stability? Would you expect a destabilization in the reporter containing a non-optimal sequence? As far as I understood, this reporter was designed based on the mosquito codon usage, what happens if it is transfected in a human cell line?

4. It is striking that the most optimal codon for Arginine in human cells is the least optimal one in mosquito cells. As a result, the DENV RSCU is not as dramatically different from the mosquito one compared to the human one. Is this because the DENV2 genome has been isolated from human samples? Is there a way to retrieve genome(s) that have been isolated from mosquitoes to compare the data?

5. The tRNA analysis has been performed only on the Arginine-decoding tRNAs, which is a bit limited. Are there some small RNA sequencing data available from DENV-infected samples, from which the authors could retrieve tRF accumulation data? This could provide a clue regarding the RNAs that are preferentially cleaved upon DENV infection.

6. Finally, how can we reconcile the authors' observation with the fact that for a number of viruses, the infection results in a global translation shutoff, which is expected to completely shift the cellular pool of ribosomes and tRNAs to a subset of mRNAs (mostly viral)?

Reviewer #3:

A very nice study claiming selection for non optimal codons in dengue virus with respect to both hosts. The results are intriguing, even if not yet understood fully (why non-optimal?). I'd ultimately be positive, but outstanding questions must be met first

1. Figure 1C shows a modest and marginally significant negative correlation between the DENV2 RSCU and that of human optimal codons. The authors take that to suggest deliberate negative correlation and selection for non-optimal codons. I reject this conclusion. The negative correlation is sufficiently low to be explained by a simple lack of selection. Genetic drift could have produced that extent of negative correlation. Further, mutational patterns could have given raise to the observed RSCU. I'd urge the authors to investigate mutational biases in the dengue virus genome replication and examine if they could have given raise to the observed codon usage with no need to invoke selection (for negative correlation)

2. Related - the authors claim that their RSCU results can't be explained by CpG di-nucleotide frequencies "unrelated with CpG dinucleotide bias" but I can't see that they showed that to be true.

3. The serotype analysis is not convincing - maybe sortypes are not sufficiently divergent to show divergence away from the pattern of selection for non-optimal codon. A negative control is needed - for example I'd like to see if selection for nucleotide triplets in the two other frames outside the reading frame does not show the same level of similarity between the serotypes (I actually predict that they might show similarity and if that's the case the authors are simply looking at strains that have not yet diverged.) I don't know what to make of the "only share ~70-75% of their amino acid" datum - is this a good enough indication for sufficient divergence? Actually Fig. 1G shows extremely narrow range of optimal/non-optimal values for the viral genes, supporting my concern.

4. But if they are right, and negative control is passed - a minor point - the statement "These results indicate that this trend in preference for non-optimal codons is conserved across DENV serotypes" is misleading in my mind. If they are correct, then the evidence is even stronger than conservation, rather they might see the result of convergent evolution - in the various serotypes, despite divergence, selection for non-optimal codons leads towards same codons being selected.

5. The authors cite a very impressive p-value ($p = 1.59e98$) (is that 1.59×10^{-98} ?) for the deviation of the viral codon usage from human. But how have they calculated this? What is the stats test? How are they getting such low p-value for a very small effect size shown in Fig 1G? my suspicion is that the factor that inflates the p-value is the large number of serotypes - 1538. But the problem is that most stats test assume in their background statistical independence of the observations - "what's the probability under the null model to get this deviation (or higher) between the viral codons and the host, assuming that each viral genome is independent from all others. But this is far from being the case in evolutionarily related strains of a virus - they are very much dependent, so that if one of them (e.g. the ancestor) deviated even minorly from the human then all of them would, simply due to common origin. This could lead to a serious interpretation error.

6. The authors compared then 483 human viruses and show for some, more convincing negative correlations with the human codon usage. If my concern above - regarding mutation patterns and drift is warranted, then maybe they should see that viruses that belong to different types (baltimore classification, e.g. RT viruses, DNA, and RNA) - that have different mutation rates - have different negative correlations. Specifically - my prediction is - the higher the mutation rate of a virus (eg. HIV?) the more negative the correlation is simply because the virus can't select for more optimal codons due to drift.

7. To support their statement "Dengue virus preferentially uses codons that are not optimal in human or mosquitoes" I suggest that the author use a dn/ds test. They might find that there's selection for synonymous sites.

8. I don't understand the point about tRNA genes being up- or down-regulated according to their codons being "denguenized" or not (I don't like the term, I must say). They only show the example of Arg tRNAs. What about the 19 others?? Even for that the p-values look marginal.

9. Figure 5 shows a very impressive experiment in which the virus was adapted to both human and mosquito cells. Codon usage is shown to have changed towards the preferred 'denguenized' codons. This is a very subtle point. There's no question that the virus has a codon usage and that it may evolve to strengthen it even further - as shown here. But I was not yet convinced that it is the non-optimality of the codon that is being selected for.

10. The discussion suggests that "We hypothesize that viruses have evolved mechanisms to tweak abundance of tRNAs" how

do they know that the virus is tweaking the host tRNA pool?

11. THE question of this study - what does the virus gain from having non optimal codons?

Response to Reviewers

Authors' Comments: We would like to express our gratitude for all the reviewers' insightful comments and suggestions. Below, in blue, you will find our responses to each of your comments along with explanations of the changes made to the manuscript. We have conducted additional analyses and experiments in response to your feedback. While we have incorporated some of these into the main text, we chose to include others only in the "Response to Reviewers" section to maintain the clarity and focus of the paper's message.

Reviewer #1:

Codons not only define the amino acid sequence of polypeptides but also affect the amount of proteins to be synthesized through multiple mechanisms. Codon optimality is the recently unveiled effects of synonymous codon choice on mRNA stability and translational efficiency, which are deeply connected to tRNA availability. Whereas codon optimality has been reported in multiple species, from bacteria to humans, how codon optimality promotes or restricts inter-species gene expression has been poorly characterized. In this study, the authors investigated the impact of codon optimality on the infection of viruses to their hosts using mosquito-borne dengue virus (DENV) as a model. The authors first compared codon usage bias in DENV and one of its host species, humans, and showed that DENV preferentially uses codons defined as nonoptimal in humans, exemplified by the arginine AGA codon. The authors then experimentally determined codon optimality in another host, mosquitos, by measuring mRNA stability genome-wide in cultured cells and showed that DENV preferentially uses codons defined as nonoptimal in mosquitoes, too, even though codon optimality differs between the two host species. To investigate the connection between DENV codon usage bias upon infection, the authors analyzed single-cell RNA-Seq data of DENV2-infected human Huh7 cells. Interestingly, human mRNAs upregulated upon DENV2 infection were enriched with codons preferentially used in the dengue virus. The upregulation of mRNAs was accompanied by the upregulation of tRNAs, including the tRNA corresponding to the dengue-enriched AGA codon. Finally, the authors analyzed the mutation rate of the dengue virus genome during in-vitro evolution through serial passaging in human and mosquito cells. The selection of synonymous mutations occurred towards DENV's preferred codons, suggesting the advantage of viral-type codon usage bias in viral infection.

Overall, this study revealed a very interesting relationship between viral codon usage, host codon optimality, and viral infection. The results suggest that the viruses take advantage of codon optimality, at least partially, by controlling the host tRNA expression for their successful infection and propagation. This study opens a new avenue for future research regarding the mechanism of tRNA expression changes by viruses and related novel therapeutic approaches. This study will be of great interest in RNA biology, viral infection and evolution, and related medical fields. The experiments were performed elegantly, the results were analyzed appropriately and described clearly, and the discussion was dense and thoughtful. With a few minor comments listed below, I fully support the publication of this exciting paper.

Authors' Comments: We appreciate your kind comments and constructive feedback. Please see below each of the responses.

Minor comments

In addition to codon optimality, several previous studies, including the one by the authors, suggested that amino acid optimality also affects mRNA stability. In this sense, I wonder if the changes in the human mRNA level upon DENV infection could be understood partly with amino acid optimality. The distinction between the codon and amino acid optimality effects could be complex. Still, the analysis of amino acid optimality might reveal the effect of viral infection on host amino acid metabolism in addition to the tRNA expression change.

Authors' Comments: We are grateful for the reviewer's insights regarding the amino acid optimality effect on mRNA stability.

As the reviewer mentioned, we have previously proposed that amino acids might affect mRNA stability in zebrafish and *Xenopus* embryos (Bazzini et al, 2016). This hypothesis was based on the observation that in zebrafish and *Xenopus* embryos we have shown that several amino acids are encoded by either optimal (Asparagine, Glycine, Glutamic acid) or non-optimal (Histidine, Cysteine, Leucine) codons (Figure 4D and 4E from (Bazzini et al, 2016)). In human cells, we have shown that only Histidine contained synonymous codons that were exclusively non-optimal (Wu et al, 2019). This is a very important concept; we think that an amino acid can be considered optimal or non-optimal only if all the synonymous codons encoding for that amino acid share the same optimality, either all optimal or all non-optimal. If an amino acid is encoded by optimal and non-optimal codons, the amino acid optimality calculated will be dominated by the optimality of the most used synonymous codon. The amino-acid stabilization coefficient (ASC), which is calculated as the Pearson correlation coefficient between mRNA stability and amino acid occurrence, is a metric used to determine amino acid optimality (Bazzini et al, 2016). For example, we observed positive ASC scores for Leucine and Isoleucine when all the synonymous codons were used for the calculation (Figure 3-figure supplement 1B from (Wu et al, 2019)). Based on this analysis, one could conclude that Leucine and Isoleucine are optimal amino acids. However, when we remove the most abundant codon from the calculation, these amino acids would be defined as non-optimal (Figure 3—figure supplement 1B from (Wu et al, 2019)). This suggests that the ASC is strongly affected by the usage of each synonymous codon. Therefore, amino acids should only be defined as optimal or non-optimal if all the synonymous codons share the same optimality (optimal or non-optimal).

However, there are few amino acids such as Histidine and potentially Serine which are encoded solely by non-optimal codons, and Glycine is largely encoded by optimal codons (Figure 3A from (Wu et al, 2019)). Thus, the ASC scores are not strongly affected after removing the most used codon (Figure 3—figure supplement 1B from (Wu et al, 2019)). Therefore, to address the reviewer's question, we explored whether genes that exhibit differential expression upon DENV infection are enriched in Histidine, Serine, or Glycine. To study this, we compared the amino acid composition between the upregulated and the downregulated genes upon DENV infection (Figure 1 for the Reviewers). Our analysis revealed that there are only a few amino acids where the difference in amino acid composition is significant, such as Tyrosine, Proline, Aspartic Acid, Glutamic Acid, Serine and Leucine. As it was previously mentioned, classifying most amino acids as optimal or non-optimal would be misleading, since the regulatory information relies specifically on the synonymous codons. Thus, we focused our analysis on Histidine, Serine, and Glycine since these are the only three amino acids that can be proposed to be optimal or non-optimal because all the synonymous codons tend to share the same optimality

trend in human (Figure 3A from (Wu et al, 2019)). There is no significant difference between the Histidine and the Glycine frequency in the up- and downregulated genes (Figure 1 for the Reviewers). Interestingly, upregulated genes upon DENV infection show significantly higher Serine usage compared to downregulated genes (Figure 1 for the Reviewers), suggesting that Serine usage might potentially be affecting the endogenous gene expression similar to what we observed with the codons.

In summary, while we found this analysis interesting, we think its inclusion in the manuscript may divert attention from the main focus, as it pertains to a single amino acid. Therefore, we have decided to refrain from including it.

Figure 1 for the Reviewers: Comparison of amino acid relative frequencies between upregulated and downregulated human genes upon dengue virus type 2 infection. p -value < 0.01, unpaired Wilcoxon test.

Page 8, bottom, "arginine tRNA decoding CGT (slightly nonoptimal) and CCG (optimal) codons." The latter codon should be CGG (Arg), not CCG (Pro).

Authors' Comments: We appreciate the reviewer identifying this typo. This is now corrected in the manuscript in page 10, line 281.

Reviewer #2:

In this manuscript, Castellano and colleagues report the intriguing observation that viruses appear to use non-optimal codons in their genome when compared to the codon usage of their host. The authors focus their analysis on Dengue virus (DENV) and show that its Relative Synonymous Codon Usage (RSCU) differs drastically than the human one. This seems to be the

case for the Arg codon, but also more globally for a large number of other codons. Interestingly, this property is conserved in all four DENV serotypes, and the authors also found a negative correlation between RSCU fold change and human codon stability coefficient for a large collection of human as well as human and invertebrate specific viruses. Then, the authors determine the RSCU in mosquito cells, which indeed allows them to show that it does exist in this specie as well, although it differs from the human one. Strikingly, DENV RSCU also differs from the mosquito's and the virus tends to use again non optimal codons compared to the mosquito usage. Finally, the authors reanalyze previously published single-cell sequencing data as well as sequencing data of DENV following serial passaging in human or mosquito cells. This allows them to draw two important conclusions, one is that human mRNAs that are upregulated upon infection are enriched in suboptimal codons, and the other is that synonymous mutations that accumulate during passaging of the virus in one given cell type tend to favor the changes to DENV preferred codons, i.e. non optimal codons. This is a thought-provoking study that has important implications for our understanding of host-virus co-evolution, in particular to understand how the pathogen evolves to go around its host defense mechanisms. It is not completely clear though in what respect the usage of non-optimal codons benefits the virus, but the presented data appear to be solid and mostly back up the authors' conclusions. There are a few points that need to be addressed to clarify some aspects of the study and/or strengthen the authors' hypothesis.

Authors' Comments: We appreciate your time in reviewing our manuscript and are committed to incorporating your constructive feedback. Please see below each of the responses.

1. While the analysis presented by the authors clearly indicates the preferred usage of sub-optimal codons by DENV, one interesting control could be to redo the same analysis by only looking at the non-coding portions of the viral genome. At the moment, the authors focused only on CDS, which perfectly makes sense, but what happens when using sequences that are not translated. The same could be done with all viruses that have been used for Figure 1H.

Authors' Comments: We appreciate the feedback and suggestions provided by the reviewer. To address this question, we have analyzed the 3' untranslated regions (3'UTR) sequence of the DENV2 genome, which spans approximately 450 nucleotides. This analysis was specifically conducted on the 3'UTR due to the comparatively limited length of the 5'UTR, which is only 97 nucleotides long and could potentially introduce bias.

To address the reviewer's question, we quantified the triplets present in the 3'UTR of DENV type 2 strain 16681. We then calculated the Relative Synonymous Codon Usage (RSCU) within this region, comparing it to the RSCU within the coding regions of the human transcriptome. Furthermore, we assessed the relationship between the RSCU fold changes and the Codon Stability Coefficient (CSC) in humans using Spearman's correlation, exactly as we had done it for the DENV coding sequence (Figure 1C). This analysis was repeated across the three potential reading frames of the 3'UTR sequence.

As expected, no significant correlation between the RSCU of the DENV 3'UTR in any of the frames and human codon optimality was observed for DENV2, as illustrated in Figure 2 for the Reviewers and in our new Figure EV1B. This result serves as a valuable negative control, underscoring the importance of the reviewer's suggestion in enriching our manuscript.

Therefore, we have added a new Figure EV1B and a new a paragraph in the manuscript including this: page 5, lines 130-134.

Figure 2 for the Reviewers: Scatterplot showing the RSCU fold change (relative to human) for DENV2 3'UTR in the three frames and human codon stability coefficient (CSC). **A.** RSCU of DENV2 in frame 1. **B.** RSCU of DENV2 in frame 2. **C.** RSCU of DENV2 in frame 3. Spearman correlation coefficients and p -values indicated, Spearman rank correlation.

2. Based on the data contained in Table EV5, it seems that the list of viruses that has been used by the authors to generalize their observations contains both RNA and DNA viruses, including viruses that are known to infect their hosts for a very long time by going into **latency (e.g. herpesviruses)** or **persistent infection (e.g. hepatitis C virus)**. One would expect that for these viruses, the codon optimality should tend to align with the one of the host. The same could be true for **retroviruses**, which integrate into the host genome. Is this something that would be worth to check?

Authors' Comments: We appreciate the reviewer's suggestion. This is exactly one of the fascinating questions that we are interested in pursuing in the future. In brief, most of the viruses analyzed tend to preferentially use non-optimal codons. However, different types of viruses can have a different codon choice, from either being close to the host optimality or to preferentially use other non-optimal codons.

In sum, we have performed the analysis suggested for RNA and DNA viruses, as well as for viruses which go into latency, persistent viruses, and retroviruses (Figure 3 for the Reviewers).

Interestingly, DNA viruses display a more negative correlation compared to RNA viruses (Figure 3A for the Reviewers) suggesting an exacerbated preference for non-optimal codons. Furthermore, persistent viruses, like latent viruses, establish long-term infection within their host. However, while latent viruses do not replicate during latency, persistent viruses actively replicate within the host. Therefore, we hypothesized that the codon choice of latent viruses would tend to align more with human codon optimality compared to that of persistent viruses. In line with this idea, several of the latent viruses analyzed displayed a more positive correlation compared to the group of persistent viruses (Figure 3B and 3C for the Reviewers). On the other hand, while some retroviruses show a slight negative or positive correlation, three of them show a strong negative correlation (Figure 3D for the Reviewers). Altogether, this data supports the hypothesis that viruses that integrate into the host genome or do not replicate actively display a codon choice that tends to align with the host optimality. However, we also identified viruses that do not follow this trend. In the future, we plan to compare the replication strategies, codon

content and gene expression patterns during infection (and ideally tRNA profiles) to dissect the potential gene regulatory properties of the viral coding content.

Finally, the reviewer’s question also brings up another interesting point. In the future, we intend to further investigate the patterns of codon choice of viruses which use different replication strategies as well as to determine the specific synonymous codons that viruses preferentially use. This is also beyond the scope of this manuscript, but our preliminary data suggests that, for example, while many viruses prefer AGA (non-optimal) to encode arginine, many others prefer AGG (also non-optimal). Elucidating viral codon preferences and relate them with the replication strategy employed by the virus and with the potential mechanism by which tRNAs might be modulated upon infection could provide valuable insights into the mechanisms underlying virus-host adaptation and viral propagation.

We intend to investigate the patterns of codon choice of viruses with different replication strategies further in the future and several other analyses will be needed to demonstrate the selection between particular non-optimal codons. Hence, to keep the focus of our original work, we prefer to not incorporate these analyses into the manuscript. Nonetheless, we extend our sincere appreciation to the reviewer for their valuable suggestions.

Figure 3 for the Reviewers: Density plot showing Spearman rank correlation between RSCU fold change (relative to human) and human CSC for human-infecting viruses. **A.** DNA and RNA viruses. **B.** Spearman rank correlations of latent viruses are indicated with labeled dashed lines. **C.** Spearman rank correlations of persistent viruses are indicated with labeled dashed lines. **D.** Spearman rank correlations of retroviruses are indicated with dashed lines.

3. Regarding the reported that has been used in Figure 2C, what is the effect of the frameshift on mRNA stability? Would you expect a destabilization in the reporter containing a non-optimal sequence? As far as I understood, this reporter was designed based on the mosquito codon usage, what happens if it is transfected in a human cell line?

Authors' Comments: To address the reviewer's question, we cloned the constructs used in Figure 2C in mosquito cells, into an expression vector for human cells. Then, we co-transfected human 293T cells with each of the GFP reporters enriched in optimal or non-optimal codons alongside mCherry as an internal control and assessed the level of protein expression using cytometry analysis. While we showed that some codons share the same regulatory properties in human and mosquito cells, others show opposite optimality (Figure 2B). Therefore, since we had originally designed these constructs based on mosquito codon optimality, we hypothesized that the fold change in the expression levels between the optimal and the non-optimal reporters was going to be lower in human cells compared to mosquito cells. Indeed, in human cells, we observed 2.2-fold difference between the reporter enriched in optimal codons with respect to the one enriched in non-optimal codons (Figure 4A for the Reviewers and New Figure EV2C). The same reporters displayed 2.9-fold change in mosquito cells (Figure 4B for the Reviewers and Figure 2C). Therefore, this result supports the idea that although the codon optimality code in human and mosquito correlate, several codons show distinct regulatory properties which impact protein expression. This also highlights the importance of investigating the codon optimality mechanism in mosquito and other host organisms to study the implication that it might have in viral infections which involve several hosts, and further understand virus-host co-evolution.

Therefore, we have edited the manuscript to include this: page 7, lines 202-205.

Figure 4 for the Reviewers: 1nt frameshift reporters designed based on mosquito codon optimality were co-transfected into **A.** *Ae. albopictus* C6/36 cells or **B.** 293T human cells with a vector encoding for mCherry as an internal control. Bar plots showing that the 1nt frameshift reporter enriched in optimal codons displayed higher GFP/mCherry fluorescence intensity than its non-optimal counterpart measured by flow cytometry analysis. Mosquito cells display a greater fold difference between the optimal and the non-optimal reporters, compared to human cells. *p*-values indicated, unpaired t-test.

4. It is striking that the most optimal codon for Arginine in human cells is the least optimal one in mosquito cells. As a result, the DENV RSCU is not as dramatically different from the mosquito one compared to the human one. Is this because the DENV2 genome has been isolated from human samples? Is there a way to retrieve genome(s) that have been isolated from mosquitoes to compare the data?

Authors' Comments: We thank the reviewer for this good observation. While we were not aware of the isolation source by the time we downloaded all the genomic sequences of DENV

from ViPR, the vast majority of sequences seem to have been isolated from human samples. DENV type 2 strain 16681 sequence was also isolated from human (Kinney et al, 1997). While it might be tempting to suggest that the viral codon bias analyzed might be dominated by the human codon optimality since the samples were isolated from humans, it is unclear for how long DENV has replicated in the human cells. However, the reviewer's question brings up a very interesting point to consider in the future. Since in nature DENV necessarily alternates between human and mosquito hosts, DENV codon usage is likely the result of a compromise to adaptation to these alternating environments. However, in the future it would be interesting to study the usage of synonymous codons in natural isolates of viruses which are able to replicate in different hosts for a longer period of time, without the need of alternating between species, and determine whether adaptation to a host for a longer period of time drives the viral codon usage towards the non-optimality of that host. In other words, if a virus replicates for a long time in one species, does it select mutations towards synonymous codons which are non-optimal in that species?

Additionally, Figure 5 from our manuscript describes the data collected from a well-designed experiment in which DENV is adapted to both human and mosquito cells and the genomic sequences collected after serial passaging the virus in each host are sequenced (Dolan et al, 2021). While the serial passages in human cells or mosquito cells cannot be compared to natural infection and the timing and nature of the experiments can have their own limitations, this experiment provides a good dataset to study the impact of two Arginine-encoding codons with opposite optimality on the infection. Our analysis of the effect of synonymous mutations on viral fitness suggests that synonymous mutations from AGG (a non-optimal codon in both human and mosquito) to CGG (non-optimal codon in mosquito and optimal codon in human) (Figure 2B) increased the viral fitness after adapting DENV to mosquito cells, while the viral fitness decreased after passaging the virus in human cells (Figure 5D). This suggests that, under these conditions, mutating AGG to a more optimal codon seems to be deleterious for the viral fitness, while mutations to a more non-optimal codon seem to be beneficial.

In sum, in the future we are planning on analyzing the usage of synonymous codons in natural isolates of viruses which replicate in different hosts for a longer period.

5. The tRNA analysis has been performed only on the Arginine-decoding tRNAs, which is a bit limited. Are there some small RNA sequencing data available from DENV-infected samples, from which the authors could retrieve tRF accumulation data? This could provide a clue regarding the RNAs that are preferentially cleaved upon DENV infection.

Authors' Comments: We appreciate the reviewer's comment and recognize the limitation of investigating only Arginine-decoding tRNAs. While to our knowledge there are not small RNA sequencing data available from DENV-infected samples, we are particularly fascinated by tRNA and tRF-mediated regulation of mRNA stability and translation during viral infections. A genome-wide tRNA quantification was beyond the scope of this study, however, we intend to explore this avenue in parallel with tRF following infection with DENV and other viruses. To explore the possibility of tRF generation and their potential regulatory role in virus infection, conducting both tRNA-seq and small-RNA-seq simultaneously will be crucial. This approach is essential because if a tRNA shows increased abundance upon virus infection, any potential tRFs generated from it may also be upregulated simply due to the higher abundance of the mature tRNA. From our perspective, the most interesting tRFs will be those that show an upregulation during viral infection which cannot be attributed to an increase in the mature tRNA population.

With our findings indicating that numerous viruses exhibit a preference for specific non-optimal synonymous codons, our next step will be to conduct tRNA-seq during the infection of a small number of viruses, including DENV. It is important to note that measuring tRNA presents challenges, and while we have done it before (Bazzini et al, 2016) and we have tried different protocols, achieving consistent library preparation has remained difficult. Once we establish the optimal method for tRNA-seq profiling in our laboratory, we will proceed to conduct the small-RNA and tRNA profiles simultaneously.

6. Finally, how can we reconcile the authors' observation with the fact that for a number of viruses, the infection results in a global translation shutoff, which is expected to completely shift the cellular pool of ribosomes and tRNAs to a subset of mRNAs (mostly viral)?

Authors' Comments: We appreciate the reviewer's comment. This is a very interesting point, because we have previously shown that the effect of codon optimality on gene expression depends on the level of translation, and therefore, on the number of ribosomes loaded onto an mRNA. Specifically, we assessed the impact of different 5' and 3' UTR sequences from mRNAs associated with different levels of translation (strong, middle, weak) based on zebrafish ribosome profiling on the mRNA level and protein expression of frameshift reporters (optimal and non-optimal) in human cells (Figure 5 from (Wu et al, 2019)). We observed that optimal and non-optimal paired reporters with high translation rates (strong 5' and 3' UTR sequences) showed greater differences at both mRNA and protein levels compared to these paired reporters with lower translation efficiencies (weak 5' and 3' UTR sequences) (Figure 5B and 5C from (Wu et al, 2019)). In sum, the impact of codon composition on mRNA stability was greater in reporters that displayed a higher level of translation, indicating that the effect of codon optimality on gene expression might depend on other cis- and trans- regulatory elements which modulate translation level in the cell. Therefore, codon-mediated effects on host gene expression might be higher upon viral infections which result in a global translation shutoff, such as herpes simplex virus type 1 (HSV-1) (Walsh et al, 2013). In agreement with this, we have previously shown that HSV-1 infection reduces the translation efficiency of endogenous mRNAs globally in human cells (Figure 5-figure supplement 1D from (Wu et al, 2019)), and the differences in mRNA levels between transcripts enriched in optimal versus non-optimal codons were reduced after infection (Figure 5D from (Wu et al, 2019)). However, when we investigated the enrichment of each codon in the group of up-regulated genes compared to downregulated genes upon DENV infection, we found that not all non-optimal codons are equally affected (Figure 4E). For instance, CAT is the most non-optimal codon in human, however it is not enriched in the group of up-regulated genes, and it is not preferentially used by DENV (Figure 4E). Specifically, within the non-optimal codons, the ones preferentially used by DENV are the most enriched in the genes upregulated. Additionally, our results indicate that the tRNAs decoding codons AGA and AGG are upregulated upon DENV infection (Figure 4F). These findings suggest that there is a codon-mediated regulation acting on specific codons during virus infection, however it is possible that the magnitude of the effect is influenced by the level of translation in the host cell under these conditions.

Reviewer #3:

A very nice study claiming selection for non optimal codons in dengue virus with respect to both hosts. The results are intriguing, even if not yet understood fully (why non-optimal?). I'd ultimately be positive, but outstanding questions must be met first

1. Figure 1C shows a modest and marginally significant negative correlation between the DENV2 RSCU and that of human optimal codons. The authors take that to suggest deliberate negative correlation and selection for non-optimal codons. I reject this conclusion. The negative correlation is sufficiently low to be explained by a simple lack of selection. Genetic drift could have produced that extent of negative correlation. Further, mutational patterns could have given raise to the observed RSCU. I'd urge the authors to investigate mutational biases in the dengue virus genome replication and examine if they could have given raise to the observed codon usage with no need to invoke selection (for negative correlation)

Authors' Comments: We appreciate the reviewer's feedback and suggestions. There are few things to mention about the reviewer concern about the not strong correlations. We have taken three complementary approaches to investigate the relationship between codons preferentially used by the virus (DENV2 RSCU) and human codon optimality.

The first approach is the one shown in the manuscript. This is a global approach analyzing all 61 codons simultaneously, as shown in Figure 1D. The advantage of this approach is that all codons can be analyzed at the same time. However, we anticipated that the correlation was going to be weak because the 61 codons are ranked based on the CSC score (from less optimal to more optimal), which does not consider each amino acid as a unit, nor the synonymous codons. If we take Proline as an example, this amino acid is encoded by one optimal codon and three slight non-optimal ones (Figure 1B). DENV preferentially uses CCA (Figure 1D). CCA is a non-optimal codon, however it is not extremely non-optimal compared to the other 60 codons (Figure 2B). Therefore, DENV RSCU analysis will show a high RSCU value for a codon that is slightly non-optimal, contributing to a weaker correlation. We think it is notable to still see a negative and significant correlation despite this limitation. Furthermore, besides DENV2 isolate 16681, our analysis of isolates including all four DENV serotypes confirmed our observation that DENV preferentially uses non-optimal codons relative to human. Finally, in response to suggestions from the previous reviewers, we analyzed the untranslated region of DENV using the same approach and as expected, we found a non-significant correlation (Figure 2 for the Reviewers). Therefore, we decided to use this approach in Figure 1 of our manuscript since it shows the big picture analyzing all synonymous codons.

In the second approach we first made two groups of codons based on the RSCU, those that are preferentially used by DENV2 ("High RSCU" = Log_2 RSCU fold change > 0.4), and those that are not preferentially used by DENV2 ("Low RSCU" = Log_2 RSCU fold change < -0.4). Then, we interrogated the CSC scores (optimality) for these two groups (Figure 5A and 5B for the Reviewers). Codons associated with high DENV2 RSCU displayed a lower median CSC score (non-optimal) compared to codons showing low DENV2 RSCU (p -value = 0.008, t-test for DENV2 isolate 16681, and p -value=0.006, t-test for all DENV2 isolates shown in Figure 1E and 1F) (Figure 5A and 5B for the Reviewers), indicating that codons preferentially used by DENV2 are non-optimal.

In the third approach, we created two groups of codons: “optimal” ($CSC > 0.03$) and “non-optimal” ($CSC < -0.03$). We then compared the median value of DENV2 RSCU relative to human for these two groups (Figure 5C and 5D for the Reviewers). Non-optimal codons showed a higher median RSCU compared to optimal codons for DENV2 isolate 16681 (p -value = 0.043, t-test) and all DENV2 isolates analyzed in Figure 1E and 1F (p -value = 0.044, t-test) (Figure 5C and 5D for the Reviewers).

These two independent analyses, together with the correlation analysis shown in Figure 1C, support the idea that codons preferred by DENV2 are more non-optimal than DENV2 unpreferred codons. However, the limitation of the last two approaches is that it does not take into consideration the impact of all 61 codons on the relationship between DENV2 RSCU and human codon optimality. Additionally, viruses might preferentially use different codons and so the comparison between viruses might be unfair.

Importantly, independently of the approach taken, we observed that hundreds of viruses preferentially use non-optimal codons relative to human and therefore, we only showed the first approach to make the paper easy to follow. In sum, based on these observations, we proposed that a non-optimal codon choice is evolutionary conserved across several viruses. Regarding genetic drift, please refer to the responses provided below.

Figure 5 for the Reviewers: Analysis of the relationship between codons preferentially used by DENV and human codon optimality. **A.** Boxplot showing the Codon Stability Coefficient (CSC) of the most preferred (High RSCU) or unpreferred (Low RSCU) codons by DENV2 strain 16681. **B.** Boxplot showing the Codon Stability Coefficient (CSC) of the most preferred (High RSCU) or unpreferred (Low RSCU) codons by DENV2 isolates. **C.** Boxplot showing the RSCU fold change of DENV2 strain 16681 relative to human for the most optimal or non-optimal codons. **D.** Boxplot showing the RSCU fold change of DENV2 isolates relative to human for the most optimal or non-optimal codons. p -values indicated, unpaired t-test.

2. Related - the authors claim that their RSCU results can't be explained by CpG di-nucleotide frequencies "unrelated with CpG dinucleotide bias" but I can't see that they showed that to be true

Authors' Comments: We appreciate the reviewer's comment. Coding sequences serve as targets for evolution operating at various levels such as nucleotide, codon, and amino acid levels. In consequence, several types of compositional bias have been identified in viral genomes, including nucleotide bias, codon bias, dinucleotide bias and codon pair bias. Disentangling the influence of these different levels on gene sequences has proven to be challenging. Codons, being nucleotide triplets, are influenced by biases at both the nucleotide and di-nucleotide levels. Additionally, codons serve as the fundamental unit of translation, specifying amino acids that

ultimately dictate protein function. Consequently, selection operates on codons, constraining the flexibility of the nucleotide sequence. Therefore, it is likely that these various forms of compositional bias collectively shape the coding sequence. Based on our current study, we propose that codon optimality might constitute an additional factor that can impact the sequence of viral genomes. However, we do not contend that codon optimality alone dictates the viral sequence. It is important for us to highlight that while codon optimality may play a role, it is not the sole determining factor. Therefore, we had also considered the potential role of CpG dinucleotide bias on viral codon choice. While the potential influence of CpG di-nucleotide avoidance on viral preference for non-optimal codons cannot be dismissed, we mentioned in the text that there are three lines of evidence suggesting that CpG alone cannot fully account for the viral codon choice:

- DENV prefers specific non-optimal codons over other synonymous codons which do not show CpG di-nucleotide in positions 1-2, 2-3 or at the codon boundaries. For example, in the text we mention that non-optimal codons unrelated with CpG dinucleotide bias such as ATA, CAA, ACA, AGA, AAA, CCA, TCA and TTA, are preferentially used by DENV over other synonymous codons which are also unrelated with CpG dinucleotide (Figure 1D).
- Viruses which only infect mosquitoes also prefer non-optimal codons even though mosquitoes lack the suppression of CpG dinucleotide observed in vertebrates and underrepresentation of CpG has not been reported in insect specific viruses (Figure 3E). In this regard, the mosquito codon optimality code defined in the present study helped to disentangle the CpG di-nucleotide bias and viral codon choice. This encourages further investigation to characterize the codon optimality mechanism across various species that serve as hosts for viruses and other pathogens.
- Relative quantification of tRNA levels showed a correlation between changes in human tRNA levels for Arginine upon DENV infection and the codons preferentially used by DENV (Figure 4F). This modulation of tRNA levels supports the idea that there might be codon-dependent changes in gene expression induced by DENV infection.
- While dsRNA and large dsDNA viruses do not show an under-representation of CpG dinucleotides in their genomes (Gaunt & Digard, 2022), our analysis of 483 virus genomes indicates that viruses including Human mastadenovirus A, Human alphaherpesvirus 3 and African swine fever virus with large dsDNA genomes display a preference towards non-optimal codons, as well as dsRNA viruses such as Mammalian orthoreovirus 3 (Table EV4).

However, we have also observed and mentioned in the manuscript that some least frequently used codons by DENV, such as GCG, CGA, CGC, CGT, CCG, TCG, and CGG (Figure EV4B), are CpG containing codons. Therefore, the potential impact of CpG di-nucleotide avoidance on viral codon choice cannot be dismissed. Altogether, evidence indicates that nucleotide, dinucleotide and codon composition are under evolutionary pressures which will act collectively to shape the sequence of viral genomes.

We have added the sentence about the dsRNA and large dsDNA viruses in the discussion: page 13, lines 391-395.

3. The serotype analysis is not convincing - maybe sortypes are not sufficiently divergent to show divergence away from the pattern of selection for non-optimal codon. A negative control is needed - for example I'd like to see if selection for nucleotide triplets in the two other frames outside the reading frame does not show the same level of similarity between the serotypes (I actually predict that they might show similarity and if that's the case the authors are simply looking at strains that have not yet diverged.) I don't know what to make of the "only share ~70-75% of their amino acid" datum - is this a good enough indication for sufficient divergence? Actually Fig. 1G shows extremely narrow range of optimal/non-optimal values for the viral genes, supporting my concern.

Authors' Comments: There are few important things to mention about the Reviewer's comment. Phylogenetic analyses have indicated that all four epidemic/endemic DENV serotypes evolved independently from sylvatic progenitors that utilize nonhuman primate hosts and *Aedes* vectors. Estimations of divergence periods suggested that the emergence of endemic or epidemic DENV likely occurred between approximately 100 and 1,500 years ago (Holmes & Twiddy, 2003; Wang et al, 2000). Therefore, the common ancestor of the four DENV serotypes likely existed long before the estimated divergence times of the endemic/epidemic and sylvatic forms of each serotype. While this does raise the possibility that selection for non-optimal codons could be attributed to convergent evolution, it also prompts the question of whether the codon optimality code is conserved between chimpanzees and humans. Codon optimality in chimpanzees has not been described yet. However, considering the observation that codon optimality exhibits significant conservation levels between humans and zebrafish (Figure 2B), and noting that the evolutionary distance between these two species exceeds that of humans and chimpanzees (Braasch et al, 2016; Chen & Li, 2001), it is possible that codon optimality in humans and chimpanzees would also highly overlap. Therefore, it is plausible that all serotypes of DENV adapted to use codons that are non-optimal in chimpanzees.

Second, we have now calculated the relationship between the RSCU and codon optimality in the 3'UTR of the four DENV serotypes, please see answer to Reviewer #2, suggestion #1 (including new Figure EV1B). In sum, we do not find a significant correlation between the RSCU in any of the 3 potential frames with human codon optimality (data not shown), suggesting that the bias is in the coding sequence. Furthermore, we have not analyzed the frame +1 and +2 from DENV coding sequence because we think the results will not be conclusive. Based on our experience working with codon optimality in human cells and zebrafish and *Xenopus* embryos, the bias observed in the correct frame impacts on the other two frames, and therefore the results will be inconclusive.

We have added a sentence about the serotypes in the discussion: page 11, lines 332-336.

4. But if they are right, and negative control is passed - a minor point - the statement "These results indicate that this trend in preference for non-optimal codons is conserved across DENV serotypes" is misleading in my mind. If they are correct, then the evidence is even stronger than conservation, rather they might see the result of convergent evolution - in the various serotypes, despite divergence, selection for non-optimal codons leads towards same codons being selected.

Authors' Comments: We appreciate the reviewer's feedback. Please refer to the response provided in Question #3 for further details. We have also modified the text as we mentioned in Question #3.

5. The authors cite a very impressive p-value ($p = 1.59e98$) (is that 1.59×10^{-98} ?) for the deviation of the viral codon usage from human. But how have they calculated this? What is the stats test? How are they getting such low p-value for a very small effect size shown in Fig 1G? my suspicion is that the factor that inflates the p-value is the large number of serotypes - 1538. But the problem is that most stats test assume in their background statistical independence of the observations - "what's the probability under the null model to get this deviation (or higher) between the viral codons and the host, assuming that each viral genome is independent from all others. But this is far from being the case in evolutionarily related strains of a virus - they are very much dependent, so that if one of them (e.g. the ancestor) deviated even minorly from the human then all of them would, simply due to common origin. This could lead to a serious interpretation error.

Authors' Comments: We thank the reviewer for the comment. Indeed, conceptually there was not needed to calculate a p-value or do any statistical test. Our goal with the analysis in Figure 1G was to address how non-optimal is the DENV genome, as a single ORF, compared to the human transcriptome and not to define whether DENV was statistical different from the transcriptome. Therefore, we agree with the reviewer in that performing a statistical test to compare these two distributions is not accurate nor informative. Therefore, we have revised the text to reflect the observation that DENV isolates exhibit a proportion of optimal/non-optimal codons below the ~34% of the human endogenous genes in page 5, lines 148-151.

6. The authors compared then 483 human viruses and show for some, more convincing negative correlations with the human codon usage. If my concern above - regarding mutation patterns and drift is warranted, then maybe they should see that viruses that belong to different types (baltimor classification, e.g. RT viruses, DNA, and RNA) - that have different mutation rates - have different negative correlations. Specifically - my prediction is - the higher the mutation rate of a virus (eg. HIV?) the more negative the correlation is simply because the virus can't select for more optimal codons due to drift.

Authors' Comments: We agree with the rationale of the comment. Therefore, we have performed the analysis for the RNA and DNA viruses separately, as well as in viruses which go into latency, persistent viruses, and retroviruses (Figure 3 for the Reviewers). Please refer to the response provided for question #2 from Reviewer #2, where we have addressed similar suggestions.

Regarding the specific comment about the drift, interestingly, DNA viruses displayed a more negative correlation compared to RNA viruses (Figure 3A for the Reviewers). This observation supports the hypothesis that viruses preferentially use non-optimal codons, and it suggests that this preference is not merely a consequence of genetic drift, given that DNA viruses typically exhibit lower mutation rates compared to RNA viruses. As the reviewers suggested, if the drift was the major driving force, then the DNA viruses should exhibit a lower correlation compared to RNA viruses and not higher. However, as we mentioned in response to question #2 from the Reviewer #2, this was just the first approach to identify that several viruses

preferentially use non-optimal codons. However, this is not implying that every virus will share the same codon choice. For example, and beyond the scope of this manuscript, our preliminary data suggests that while many viruses prefer AGA (non-optimal) to encode arginine, many others prefer AGG (non-optimal). Moreover, viruses with different infection dynamics and replication strategies might evolve their coding sequence differently. This an intriguing aspect that we intend to pursue further in the future.

7. To support their statement "Dengue virus preferentially uses codons that are not optimal in human or mosquitoes" I suggest that the author use a dn/ds test. They might find that there's selection for synonymous sites.

Authors' Comments: We thank the reviewer for the comment. It is certainly an interesting point to investigate. The dN/dS test assumes that synonymous codons are neutral because they do not change the encoded amino-acid and uses the synonymous rate as a reference point to determine whether fixation of nonsynonymous mutations in the population is influenced by natural selection acting on the protein. However, since we have shown that synonymous substitutions are not silent from a regulatory point of view, and our goal would be to study whether natural selection is acting specifically on synonymous sites, we think that the dN/dS test will not be very informative to address this question. An adequate analysis for natural selection of synonymous sites and its relationship with codon optimality is out of the scope of this project, but we intend to further investigate evolution under codon optimality constraints in future projects.

8. I don't understand the point about tRNA genes being up- or down-regulated according to their codons being "denguenized" or not (I don't like the term, I must say). They only show the example of Arg tRNAs. What about the 19 others?? Even for that the p-values look marginal.

Authors' Comments: The rationale to study tRNA level during viral infection was the following: Viral RNA translation is strictly dependent on the host cell resources, including tRNAs. During the process of translation, each codon will be decoded by its cognate tRNA. It has been shown that the tRNAs decoding optimal codons are highly expressed, while the ones decoding non-optimal codons tend to be lowly expressed (Bazzini et al, 2016; Despic & Neugebauer, 2018; Frumkin et al, 2018; Richter & Collier, 2015; Wu et al, 2019). Moreover, optimal codons are often decoded by tRNA that display a higher ratio of charged (with amino acid) to total tRNA whereas non-optimal codons tend to be associated with lower ratios of charged to total tRNA (Wu & Bazzini, 2023; Wu et al, 2019). Therefore, it was proposed that the optimality of a given codon might be dictated by the tRNA ready to be used by the ribosome (Rak et al, 2018; Wu & Bazzini, 2023). Moreover, the tRNA repertoire is dynamic, and it has been shown that cells can alter their tRNA abundance to selectively regulate protein synthesis during stress conditions (Torrent et al, 2018). For instance, proliferation- and differentiation-related genes show a distinct codon usage, and each state upregulates the tRNA set whose codons are enriched in its transcriptome (Gingold et al, 2014). Further, upregulation of specific tRNAs drives metastasis by enhancing stability and translation of transcripts enriched in their cognate codons (Goodarzi et al, 2016).

Our observations in this study indicate that DENV prefers non-optimal codons ('denguenized') relative to humans. We reasoned that tRNAs decoding "denguenized" codons

will be in high demand upon DENV infection, to allow successful translation and propagation of the virus. Thus, we interrogated whether the tRNA level could change upon virus infection.

Measuring tRNA levels genome-wide poses significant challenges and often lacks reproducibility, so we focused on measuring the relative abundance of tRNAs decoding Arginine codons using qPCR for the following reasons:

- The analysis of DENV codon choice revealed that DENV prefers AGA (non-optimal) and avoids CGG (optimal).
- tRNA TGT – decoding codon AGA- was shown to be regulable in other biological conditions (cancer) (Orellana et al, 2021)
- The publicly available sequence of oligos for amplifying arginine tRNAs (Hsu et al, 2023) ensured the prevention of any technical issues associated with the qPCR.

Interestingly, the relative abundance of arginine tRNA decoding AGA was significantly upregulated in human cells infected with DENV compared to mock infected cells, while CGG (“not-denguenized”/optimal codon) was not, supporting our hypothesis of modulation of tRNA levels upon infection. We recognize the limitation of investigating only Arginine-decoding tRNAs. As we mentioned to Reviewer #2, a genome-wide tRNA quantification was beyond the scope of our study. However, we are fascinated by tRNA as well as for tRNA fragments-mediated regulation of mRNA stability and translation during viral infections and we intend to explore this avenue in the future using different viruses.

9. Figure 5 shows a very impressive experiment in which the virus was adapted to both human and mosquito cells. Codon usage is shown to have changed towards the preferred 'denguenized' codons. This is a very subtle point. There's no question that the virus has a codon usage and that it may evolve to strengthen it even further - as shown here. But I was not yet convinced that it is the non-optimality of the codon that is being selected for.

Authors' Comments: It is likely that the viral coding sequence is influenced by selection pressures arising from a range of factors, including RNA structure, translation elongation, protein folding, and functional requirements. Selection operates on the sequence at various levels, including nucleotides, dinucleotides, codons, and amino acids.

In the experiment in which we serially passaged DENV in human or mosquito cells, we observed that synonymous mutations towards ‘denguenized’ codons showed higher “fitness” than mutations towards ‘non-denguenized’ codons (Figure 5E), suggesting that the non-optimal codons are beneficial for viral fitness compared to changing to an optimal codon. However, this codon selection might also be driven by nucleotides, local RNA structures in the viral genome, and other features.

In our view, it is remarkable to see this result in this type of experiment because it suggests that non-optimal codons tend to show regulatory properties that are advantageous for the virus, while usually human mRNAs enriched in these codons show low abundance and reduced level of translation.

Based on our observation, we think it would be ideal to perform a similar experiment but in opposite direction: to generate several viral sequences with several synonymous codon mutations, for example some optimized (enriched in optimal codon), some deoptimized and some with the same optimality but different codon composition. Then, we would monitor the

synonymous mutations after serial passaging the virus in human or mosquito cells, similar to the experiment shown in Figure 5 but starting with a complex viral population rather than a single sequence. If the non-optimality of the codons is a main driver of the viral codon usage during virus-host adaptation, we would expect the viral population to accumulate synonymous mutations encoded by non-optimal codons over time. However, it is very important to mention that this experiment poses challenges, particularly due to the implications of recoding the viral genome on public health, and we are unable to conduct it under the existing biosafety conditions.

10. The discussion suggests that "We hypothesize that viruses have evolved mechanisms to tweak abundance of tRNAs" how do they know that the virus is tweaking the host tRNA pool?

Authors' Comments: We appreciate the comment, as it raises an important point. The modulation of tRNA abundance could indeed be attributed to viral manipulation, but it is also possible that this phenomenon is a consequence of the host's response to infection. We hypothesize that the abundance of tRNAs might be modulated upon virus infection, leading to changes in gene expression in the host cell. However, the host-pathogen interaction is often a coevolution, so it is possible that the virus and/or the host cell might induce the regulation of tRNAs upon infection. Our work, which demonstrates the modulation of the abundance of tRNAs decoding arginine, lays the groundwork for future investigations into the potential regulation of other tRNAs and the molecular mechanisms that drive these changes. One plausible scenario is that viruses have evolved mechanisms to tweak abundance of tRNAs decoding non-optimal codons to misregulate expression of host genes and favor expression of viral genes. In agreement with this, it has been recently reported that Chikungunya infection induces a codon-specific reprogramming of the host translation machinery to favor viral protein expression by regulating tRNA modifications (Jungfleisch et al, 2022). Alternatively, modulations of the tRNA pool might occur as a consequence of the cellular response to infection, and viruses might have adapted their codon usage to effectively propagate in this environment. Regardless, our hypothesis proposes that the preference for non-optimal codons confers an advantage to the virus, as this is a trend observed in hundreds of human viruses. In sum, we thank the reviewer for the comments, and we have added this suggestion in the discussion page 13, lines 397-401.

11. THE question of this study - what does the virus gain from having non optimal codons?

Authors' Comments: This is the main biological question raised from our work to address in the future. While further research is necessary to address this question, our data provides the foundational work to explore the mechanism by which non-optimal codons might be advantageous for viral protein translation. It is fascinating that the preference for non-optimal codons is conserved across hundreds of human viruses, including both DNA and RNA viruses. Our observation that viruses which only infect humans or humans and vertebrates showed a stronger negative correlation than viruses which infect humans and invertebrates, such as mosquito-borne viruses, suggests that selective constraints imposed by divergent hosts might pose an evolutionary pressure that may shape the codon preference. Additionally, while our data supports the hypothesis that viruses that integrate into the host genome or do not replicate actively display a codon choice that tends to align with the host optimality, we also identified viruses that do not follow this trend. Future investigations focusing on the codon choice of viruses which use different replication strategies could shed light on the mechanism by which a

non-optimal codon usage might be advantageous to the virus. Further, our preliminary data not shown in this manuscript, suggests that viruses preferentially use specific non-optimal synonymous codons. Profiling and quantification of tRNAs and tRNA-derived fragments upon virus infection could help elucidate the potential mechanism by which tRNAs might be modulated upon virus infection and the influence they ultimately have on viral translation and fitness. It would also be interesting to study whether the regulation of tRNA levels, tRNA modifications and tRNA-derived fragments are strategies directly employed by the virus to optimize the host environment for viral translation or whether it is part of the host cell response to the infection and the virus has adapted to this environment. Finally, we are currently investigating the factors that regulate codon optimality in human cells. Our forthcoming research aims to elucidate the impact of these regulators on viral translation and fitness through depletion or overexpression experiments.

In sum, future investigations into the mechanisms governing changes in codon optimality following virus infection, along with the potential advantages conferred to viral fitness, could provide valuable insights into the intricate dynamics of virus-host coevolution.

References

Bazzini AA, Del Viso F, Moreno-Mateos MA, Johnstone TG, Vejnar CE, Qin Y, Yao J, Khokha MK, Giraldez AJ (2016) Codon identity regulates mRNA stability and translation efficiency during the maternal-to-zygotic transition. *The EMBO journal* **35**: 2087-2103

Braasch I, Gehrke AR, Smith JJ, Kawasaki K, Manousaki T, Pasquier J, Amores A, Desvignes T, Batzel P, Catchen J, Berlin AM, Campbell MS, Barrell D, Martin KJ, Mulley JF, Ravi V, Lee AP, Nakamura T, Chalopin D, Fan S et al (2016) The spotted gar genome illuminates vertebrate evolution and facilitates human-teleost comparisons. *Nat Genet* **48**: 427-437

Chen FC, Li WH (2001) Genomic divergences between humans and other hominoids and the effective population size of the common ancestor of humans and chimpanzees. *Am J Hum Genet* **68**: 444-456

Despic V, Neugebauer KM (2018) RNA tales - how embryos read and discard messages from mom. *J Cell Sci* **131**

Dolan PT, Taguwa S, Rangel MA, Acevedo A, Hagai T, Andino R, Frydman J (2021) Principles of dengue virus evolvability derived from genotype-fitness maps in human and mosquito cells. *eLife* **10**

Frumkin I, Lajoie MJ, Gregg CJ, Hornung G, Church GM, Pilpel Y (2018) Codon usage of highly expressed genes affects proteome-wide translation efficiency. *Proc Natl Acad Sci U S A* **115**: E4940-e4949

Gaunt ER, Digard P (2022) Compositional biases in RNA viruses: Causes, consequences and applications. *Wiley Interdiscip Rev RNA* **13**: e1679

Gingold H, Tehler D, Christoffersen NR, Nielsen MM, Asmar F, Kooistra SM, Christophersen NS, Christensen LL, Borre M, Sørensen KD, Andersen LD, Andersen CL, Hulleman E, Wurdinger T, Ralfkiær E, Helin K, Grønæk K, Ørntoft T, Waszak SM, Dahan O et al (2014) A dual program for translation regulation in cellular proliferation and differentiation. *Cell* **158**: 1281-1292

Goodarzi H, Nguyen HCB, Zhang S, Dill BD, Molina H, Tavazoie SF (2016) Modulated Expression of Specific tRNAs Drives Gene Expression and Cancer Progression. *Cell* **165**: 1416-1427

Holmes EC, Twiddy SS (2003) The origin, emergence and evolutionary genetics of dengue virus. *Infect Genet Evol* **3**: 19-28

Hsu DJ, Gao J, Yamaguchi N, Pinzaru A, Wu Q, Mandayam N, Liberti M, Heissel S, Alwaseem H, Tavazoie S, Tavazoie SF (2023) Arginine limitation drives a directed codon-dependent DNA sequence evolution response in colorectal cancer cells. *Sci Adv* **9**: eade9120

Jungfleisch J, Böttcher R, Talló-Parra M, Pérez-Vilaró G, Merits A, Novoa EM, Díez J (2022) CHIKV infection reprograms codon optimality to favor viral RNA translation by altering the tRNA epitranscriptome. *Nat Commun* **13**: 4725

Kinney RM, Butrapet S, Chang GJ, Tsuchiya KR, Roehrig JT, Bhamarapavati N, Gubler DJ (1997) Construction of infectious cDNA clones for dengue 2 virus: strain 16681 and its attenuated vaccine derivative, strain PDK-53. *Virology* **230**: 300-308

Orellana EA, Liu Q, Yankova E, Pirouz M, De Braekeleer E, Zhang W, Lim J, Aspris D, Sendinc E, Garyfallos DA, Gu M, Ali R, Gutierrez A, Mikutis S, Bernardes GJL, Fischer ES, Bradley A, Vassiliou GS, Slack FJ, Tzelepis K et al (2021) METTL1-mediated m(7)G modification of Arg-TCT tRNA drives oncogenic transformation. *Molecular cell* **81**: 3323-3338.e3314

Rak R, Dahan O, Pilpel Y (2018) Repertoires of tRNAs: The Couplers of Genomics and Proteomics. *Annu Rev Cell Dev Biol* **34**: 239-264

Richter JD, Collier J (2015) Pausing on Polyribosomes: Make Way for Elongation in Translational Control. *Cell* **163**: 292-300

Torrent M, Chalancon G, de Groot NS, Wuster A, Madan Babu M (2018) Cells alter their tRNA abundance to selectively regulate protein synthesis during stress conditions. *Sci Signal* **11**

Walsh D, Mathews MB, Mohr I (2013) Tinkering with translation: protein synthesis in virus-infected cells. *Cold Spring Harb Perspect Biol* **5**: a012351

Wang E, Ni H, Xu R, Barrett AD, Watowich SJ, Gubler DJ, Weaver SC (2000) Evolutionary relationships of endemic/epidemic and sylvatic dengue viruses. *J Virol* **74**: 3227-3234

Wu Q, Bazzini AA (2023) Translation and mRNA Stability Control. *Annual review of biochemistry*

Wu Q, Medina SG, Kushawah G, DeVore ML, Castellano LA, Hand JM, Wright M, Bazzini AA (2019) Translation affects mRNA stability in a codon-dependent manner in human cells. *eLife* **8**

3rd Jun 2024

Manuscript Number: MSB-2024-12244R

Title: Dengue virus preferentially uses human and mosquito non-optimal codons

Dear Ariel,

Thank you for sending us your revised manuscript. We have now heard back from the two reviewers who were asked to evaluate your revised study. As you will see below, the reviewers think that the performed revisions have addressed most of their major concerns. However, reviewer #2 lists a remaining concern, which we would ask you to address in a revision. We would also ask you to address some minor editorial issues listed below. I would also like to inform you that I am leaving MSB, taking over a new position at EMBO, so one of my colleagues will take over handling your manuscript.

The editorial issues that need to be addressed are the following:

- Our data editors have noticed some unclear or missing information in the figure legends, please address the following:
 - Please provide the exact p values in the legends of figures 1e; 4f; 5c; EV 2b.
 - The box plots need to be defined in terms of minima, maxima, centre, bounds of box and whiskers, and percentile in the legends of figures 1e; 5e; EV 3a.
 - Please provide information related to n in the legend of figure EV 2c.
 - Please describe the nature of entity for 'n' (biological? technical?) in the legends of figures 5d-e.
 - Please define the error bars in the legend of figure EV 2c.
 - The measure of center for the error bar needs to be defined in the legend of figure 5d.
 - Please include a numbered scale bar for the heatmap present in figure EV 1a.
- Please include 5 keywords.
- The References should be formatted according to the Molecular Systems Biology reference style (i.e., ordered alphabetically and listing the first 10 authors followed by et al).
- As I mentioned in our previous correspondence, we typically do not allow data deposition at institute repositories. We would ask you to deposit the data that is currently at the Stowers Original Data Repository either at Biostudies <https://www.ebi.ac.uk/biostudies/> or to provide them as EV Datasets.
- Please remove the 'Authors Contributions' from the manuscript. The 'Author Contributions' section is replaced by the CRediT contributor roles taxonomy to specify the contributions of each author in the journal submission system. Please use the free text box in the 'author information' section of the online submission system to provide more detailed descriptions if needed (e.g., 'X provided intracellular Ca⁺⁺ measurements in fig Y').
- The figures should be called out in numerical order. Currently, Figure 3 is called out before panels of Figure 2, this needs to be corrected.
- The section order should be as follows: title page with complete author information, abstract, keywords, introduction, results, discussion, methods, data availability section, acknowledgements, disclosure and competing interests statement, references, main figure legends, tables, expanded figure legends.

Please resubmit your revised manuscript online, with a covering letter listing amendments and responses to each point raised by the referees. Please resubmit the paper ****within one month**** and ideally as soon as possible. If we do not receive the revised manuscript within this time period, the file might be closed and any subsequent resubmission would be treated as a new manuscript. Please use the Manuscript Number (above) in all correspondence.

Click on the link below to submit your revised paper.

Kind regards,

Maria

Maria Polychronidou, PhD
Senior Editor
Molecular Systems Biology

If you do choose to resubmit, please click on the link below to submit the revision online before 3rd Jul 2024.

IMPORTANT:

Please note that corresponding authors are required to supply an ORCID ID for their name upon submission of a revised manuscript (EMBO Press signed a joint statement to encourage ORCID adoption).

(<https://www.embopress.org/page/journal/17444292/authorguide#editorialprocess>)

Currently, our records indicate that the ORCID for your account is 0000-0002-2251-5174.

Link Not Available

*** PLEASE NOTE *** As part of the EMBO Press transparent editorial process initiative (see our Editorial at <https://dx.doi.org/10.1038/msb.2010.72> , Molecular Systems Biology will publish online a Review Process File to accompany accepted manuscripts. When preparing your letter of response, please be aware that in the event of acceptance, your cover letter/point-by-point document will be included as part of this File, which will be available to the scientific community. More information about this initiative is available in our Instructions to Authors. If you have any questions about this initiative, please contact the editorial office (msb@embo.org).

Reviewer #2:

The authors have addressed all of my comments. I am satisfied with this revised version. However, I feel that point 3 of Reviewer 3, which aligns with the first comment of my previous review, has not been sufficiently taken into account. Even if the authors think that it is not worth performing the analysis in frame +1 and +2 from DENV coding sequence, it would be informative to do it and discuss the results appropriately.

Reviewer #3:

The authors have address most of my comments. I was particularly intrigued by the observation that DNA viruses displayed a more negative correlation compared to RNA viruses, that addresses the issue of mutational bias as an alternative. I recommend publication. Tzachi Pilpel

Response to Reviewers

Reviewer #2:

The authors have addressed all of my comments. I am satisfied with this revised version. However, I feel that point 3 of Reviewer 3, which aligns with the first comment of my previous review, has not been sufficiently taken into account. Even if the authors think that it is not worth performing the analysis in frame +1 and +2 from DENV coding sequence, it would be informative to do it and discuss the results appropriately.

Authors' Comments:

We have analyzed the frame +1 and +2 from DENV2 coding sequence and we observed a significant negative correlation in these frames (Figure 6 for the Reviewers, WT). To study whether the bias observed in frame 1 can introduce a bias in the other two frames (+1, +2), we shuffled the codons in the coding sequence of DENV2 three times and repeated the analysis in the three frames (Figure 6 for the Reviewers, Shuffle 1, Shuffle 2, and Shuffle 3). As expected, the exact same bias was observed in DENV2 the wildtype sequence and the three shuffled sequences in frame 1 (Figure 6A for the Reviewers). Interestingly, a significant negative correlation was observed in frame 2 and frame 3 in all three shuffled sequences (Figure 6B and 6C for the Reviewers), suggesting that the sequence in frame 1 can introduce bias into frame 2 and frame 3 independent of the order of codons in frame 1. This suggests that the negative correlations in the out of frame sequences are unlikely due to selective pressures, but rather are more likely attributable to biases inherent in the in-frame sequence.

Figure 6 for the Reviewers: Scatterplot showing the RSCU fold change (relative to human) for DENV2 (WT) sequence and dengue 2 coding sequence shuffled three times (Shuffle 1, Shuffle 2, and Shuffle 3) in the three possible open reading frames and human codon stability coefficient (CSC). **A.** RSCU of DENV2 WT coding sequence or shuffled in frame 1. **B.** RSCU of DENV2 WT coding sequence or shuffled in frame 2. **C.** RSCU of DENV2 WT coding sequence or shuffled in frame 3. Spearman correlation coefficients and p -values indicated, Spearman rank correlation.

Minor changes in some figures:

In response to the reviewers' feedback during the first round, we calculated the RSCU of DENV2 3'UTR (Figure 2 for Reviewers and Figure EV1C). During this revision round, we realized that we had forgotten to exclude the STOP codons and the amino acids methionine (ATG) and tryptophan (TGG) which are encoded by a single codon and therefore their RSCU will show a value of 1. Therefore, we repeated the RSCU calculation as we previously have done it in the original manuscript excluding these codons (Figure 2 for the Reviewers and Figure EV1C provided in this submission). The conclusions remain the same after repeating this analysis. Additionally, when we included this new analysis in the previous round of revisions, we had forgotten to include the source of dengue virus 2 3'UTR in the methods section. We have edited the methods to include this (page 19, lines 588-590).

During this round of revisions, we identified some tables in which the number of codons was not calculated in the same way. This led to almost indistinguishable RSCU fold change calculation between some of the figures. The conclusions are the same after replotting the data. However, some values are slightly different compared to the previous values. We have revised the manuscript, figures, Dataset EV1, Dataset EV4, Dataset EV6 and this document to reflect the accurate values.

The modifications are nearly imperceptible, but please find below the original ("Old") and "new" figures side by side. We apologize for this minor discrepancy, but we want to maintain transparency regarding the changes made.

Fig 1C

Fig EV1B

Figure 1C: Old and new panels after revisions, including new panels in Figure EV1.

Since we have added a new panel displaying the correlation between the RSCU of DENV1, DENV3 and DENV4 and human codon stability coefficient (Figure EV1B), we have also added the legend for this panel.

Fig 1D

Figure 1D: Old and new panel after revisions.

Figure 1H: Old and new panel after revisions.

Figure 3C: Old and new panel after revisions.

Figure 3D: Old and new panel after revisions.

Fig 3E

E

Figure 3E: Old and new panel after revisions.

Fig 3F

Figure 3F: Old and new panel after revisions.

Fig EV1C
Fig 2 for the Reviewers

Figure EV1C / Figure 2 for the Reviewers: Old and new panel after revisions.

Figure 3A for the Reviewers: Old and new panel after revisions.

Fig 3 for the Reviewers

Figure 3B for the Reviewers: Old and new panel after revisions.

Fig 3 for the Reviewers

Figure 3C for the Reviewers: Old and new panel after revisions.

Fig 3 for the Reviewers

Figure 3D for the Reviewers: Old and new panel after revisions.

Fig 5 for the Reviewers

Figure 5 for the Reviewers: Old and new panels after revision.

26th Jun 2024

Manuscript number: MSB-2024-12244RR

Title: Dengue virus preferentially uses human and mosquito non-optimal codons

Dear Dr. Bazzini,

Thank you again for sending us your revised manuscript. We are now satisfied with the modifications made and I am pleased to inform you that your paper has been accepted for publication.

Kind regards,
Jingyi

Jingyi Hou, PhD
Scientific Editor
Molecular Systems Biology
